# Modelling novelty detection in the thalamocortical loop

**Chao Han**[1], **Gwendolyn English**[2,3], **Hannes P. Saal**[4], **Giacomo Indiveri**[2,3],
**Aditya Gilra**[1,5], **Wolfger von der Behrens**[2,3☯], **Eleni Vasilaki**[1,2☯] *

**1** Department of Computer Science, University of Sheffield, Sheffield, United Kingdom, **2** Institute of Neuroinformatics, University of Zurich and ETH Zurich, Switzerland, **3** ZNZ Neuroscience Center Zurich, ETH Zurich and University of Zurich, Switzerland, **4** Department of Psychology, University of Sheffield, Sheffield, United Kingdom, **5** Machine Learning Group, Centrum Wiskunde & Informatica, Amsterdam, The Netherlands

☯ These authors contributed equally to this work.
* e.vasilaki@sheffield.ac.uk

**Data Availability Statement:** The source code for running the network model and visualizing the results presented in the paper can be accessed at https://github.com/ChaoHan-UoS/SomatosensorySSA_Model.

## Abstract

In complex natural environments, sensory systems are constantly exposed to a large stream of inputs. Novel or rare stimuli, which are often associated with behaviorally important events, are typically processed differently than the steady sensory background, which has less relevance. Neural signatures of such differential processing, commonly referred to as novelty detection, have been identified on the level of EEG recordings as mismatch negativity (MMN) and on the level of single neurons as stimulus-specific adaptation (SSA). Here, we propose a multi-scale recurrent network with synaptic depression to explain how novelty detection can arise in the whisker-related part of the somatosensory thalamocortical loop. The "minimalistic" architecture and dynamics of the model presume that neurons in cortical layer 6 adapt, via synaptic depression, specifically to a frequently presented stimulus, resulting in reduced population activity in the corresponding cortical column when compared with the population activity evoked by a rare stimulus. This difference in population activity is then projected from the cortex to the thalamus and amplified through the interaction between neurons of the primary and reticular nuclei of the thalamus, resulting in rhythmic oscillations. These differentially activated thalamic oscillations are forwarded to cortical layer 4 as a late secondary response that is specific to rare stimuli that violate a particular stimulus pattern. Model results show a strong analogy between this late single neuron activity and EEG-based mismatch negativity in terms of their common sensitivity to presentation context and timescales of response latency, as observed experimentally. Our results indicate that adaptation in L6 can establish the thalamocortical dynamics that produce signatures of SSA and MMN and suggest a mechanistic model of novelty detection that could generalize to other sensory modalities.

## Author summary

Cortical sensory neurons have been shown to be capable of novelty detection, that is they respond more vigorously when a novel, unexpected stimulus is presented, and less so

**Funding:** EV acknowledges support from the Wellcome Trust Grant 110149/Z/15/Z (https://wellcome.org/). WVDB acknowledges support from the Swiss National Science Foundation, project grant no. 310030_172962. The funders had no role in study design, data collection and analysis, decision to publish, or preparation of the manuscript.

**Competing interests:** No.

when the stimulus is part of a predictable sequence. However, the neural mechanism underlying this capability is not yet fully understood. Here, we developed a minimalistic thalamocortical network model that accounts for novelty detection and reproduces physiologically observed neural response patterns in the anaesthetized somatosensory cortex. Specifically, our results demonstrate that the novelty signal arises from the recurrent interplay between thalamic neurons and cortical neurons in layers 4 and 6, without the need of other components. This work therefore provides a concrete mechanism that can serve as a starting point for further investigating the neural circuit mechanisms underlying novelty detection.

## Introduction

Sensory cortices are adept at identifying regularities and patterns obscured in the ever-changing input stream to dynamically generate predictions for the forthcoming stimulus. Their responses are then updated according to differences between the anticipated sensory inputs and the actual ones [1]. Mismatch negativity (MMN), observed in human electroencephalograph (EEG) recordings [2], and stimulus-specific adaptation (SSA), evident in single-cell recordings in animals [3], are two prominent instances of this type of short-term plasticity acting over timescales of seconds to minutes. For example, when consecutively presented with an identical stimulus over a short period, the cortical sensory neuron will decline/adapt its response and predict/expect another same one for the next stimulus. If the next stimulus violates the expectation, a larger/unadapted response will be evoked than by the expected stimulus. Therefore, both SSA and MMN conceptually represent a form of novelty detection: unexpected or infrequent stimuli are made more salient by eliciting stronger responses than expected or frequent ones.

The phenomenon of attenuated neuronal responses to common (standard) stimuli without generalizing to other, rare (deviant) stimuli has been a long-standing topic of interest, especially in the auditory system, which is referred to as SSA [4–8]. It is typically tested using the oddball paradigm that contains a repetitive standard sequence occasionally interrupted by a deviant [9]. SSA is a robust and widespread phenomenon found in multiple sensory modalities. Ulanovsky et al [4] first observed this phenomenon in the auditory cortex, and following this pioneering work, SSA has also been identified at subcortical stages of the auditory pathway such as the thalamus [10–12] and the inferior colliculus [13, 14], as well as in other sensory systems such as the somatosensory cortex [5, 8] and the visual system [15, 16]. Furthermore, SSA was demonstrated in the auditory cortex of anaesthetized [6, 7], awake [17] and freely moving rats [18], indicating that this phenomenon is not significantly affected by brain state and likely a hardwired component of sensory processing.

MMN, like SSA, is usually studied by applying the oddball paradigm and manifested in human EEG recordings as an additional negative deviation elicited by a deviant stimulus that breaks the regularity established by repeated presentations of the standard [9]. Both cortical SSA and MMN demonstrate a "true" deviance-detecting property, where the deviant response is dependent on its presentation context and beyond mere sensory input depression by applying several control paradigms [6, 8, 19]. In addition to SSA, animal models also show MMN-like responses that are homologous with human MMN in terms of their electrophysiological, pharmacological and functional properties [20, 21]. Despite significant similarities, SSA is unlikely to be the direct neuronal substrate of MMN, but may be one of the several mechanisms leading to the generation of MMN [22]. While MMN has been pharmacologically

demonstrated to be NMDA-receptor dependent [23], SSA seems to be not affected by NMDA-receptor antagonists [24]. Besides, the response latency of SSA (peaks <10 ms after stimuli onset) is substantially shorter than MMN (peaks ∼150–200 ms post-stimulus). Instead, SSA temporally matches better another event-related potential called mid-latency response (MLR) that peaks around 20 ms after stimuli onset [21]. MLR observed in humans and animals also demonstrates true deviance detection [25] and its insensitivity to NMDA-receptor antagonists [26]. Thus, it seems more plausible to postulate that SSA is the potential neuronal correlate of mid-latency response. Moreover, Musall et al. [8] recently found in the primary somatosensory cortex (S1) of anaesthetized rats a late deviant-sensitive sensory response occurring hundreds of milliseconds post-stimulus, similar to MMN latency. The late sensory response has also been tested to be context-specific. These evidence strongly implied that the late response possibly lies in the physiological origin of MMN.

Over the past decade, a large number of computational models have been proposed to account for cortical SSA, with a focus on the auditory pathway. One simple type of SSA model is based on a feedforward network with synaptic depression [6, 7, 27–29] which, unlike general neuronal fatigue, is stimulus-specific. Prominent synaptic depression is found in the thalamocortical pathway, and thus cortical SSA emerges due to the differentially adapted thalamocortical inputs [6, 27]. In this class of models, the standard's regularity cannot affect the deviant response because the responses to each stimulus feature solely depend on the amount of adaptation load accumulated in its input channel. Hence the model fails to exhibit true deviance detection in which the presentation context formed by the standards also contributes to the deviant response. Nevertheless, this flaw of the model can be addressed through some modifications, such as employing the model in a multi-layer configuration [29] or assuming the width of the tuning curve of the input channel is history-dependent [3]. On top of that, recurrent networks have been considered an alternative way to generate cortical SSA [30–34]. Predictive coding based neuronal network of the auditory cortex was studied by Wacongne et al. to account for deviance detection in the context of MMN [30]. Yarden et al. proposed a generic SSA mechanism mediated via the propagation of synchronous population activity across a local neural circuit [33]. Recently a laminar network of morphologically plausible, multi-compartmental neuron units was designed to capture SSA in the form of local field potentials [34].

However, there is still no computational model accounting for the biphasic sensory responses in S1 that can be either sensitive to the rarity and presentation context of stimuli or not, depending on the response latency [8]. In particular, the latency of the secondary cortical response cannot be accounted for by the dynamics of a recurrent network involving short-term synaptic depression that is commonly regarded as a indispensable constituent of SSA models. Thus the latency is possibly produced by some subcortical source(s) other than the intra-cortical dynamics. Inspired by the idea that SSA arises from the interaction between differentially adapted populations of neurons tuned for specific feature stimuli [33], here we developed a minimalistic hierarchal thalamocortical model for the whisker-related region of S1 (also known as the barrel cortex) with multiple timescales of synaptic and neuronal dynamics to capture the biphasic activity that demonstrates both signatures of SSA and MMN. On the short timescale of the early response, a vital element of the transient dynamics of the cortical circuitry is the so-called population spike (PS), characterized by nearly coincident firings of a pool of neurons evoked by a brief stimulus. On the longer timescale, we hypothesize that the late rhythmic cortical response is inherited from the thalamic oscillation initiated by the interplay between the thalamic reticular (RE) and thalamocortical (TC) relay cells. Our minimalistic somatosensory thalamocortical network retains only those crucial neurobiological features, such as the laminar architecture and somatotopic organization of the barrel cortex, synaptic connectivity between the barrel cortex and its corresponding thalamic nuclei, as well as

different synaptic receptors within thalamic nuclei, that are sufficient to elicit the late context-dependent deviant responses. Finally, we test the cortical capacity for predictive coding using more complex stimulus sequences. The results of our model indicate that the precision of expectation generated by sequence history impacts the neural response to stimuli that interrupt those expectations.

## Results

The architecture of the somatosensory thalamocortical network is organized in a loop as shown in Fig 1A. Based on the stereotypic somatotopic map for representing rodent whiskers, the barrel cortex is modeled as a grid of interconnected vertical columns, each of which primarily mediates stimuli from its principal whisker. Also, we only focus on neuronal clusters within layers 4 (barrels) and 6 (infrabarrels) of each cortical column, which respectively act as the afferent and efferent layers of the cortex from and to the thalamus [35–37]. The barreloids in the ventral posterior medial (VPM) nucleus of the thalamus show an identical layout to that of the barrels. Each barreloid relays the information from its principal whisker through an ascending sensory pathway to L4 of its principal as well as surrounding barrels, with stronger coupling to the principal than the surrounding ones. We adopt a minimalistic structure of the thalamocortical network by limiting our model to the most important "ingredients", meaning VPM, TRN and barrel cortex layers 4 and 6, while leaving out other important structures and layers. Hence, we just add a direct shortcut connection from L4 to L6 within every cortical column, which is sufficient in our model to demonstrate cortical SSA, although in reality the flow of excitation from L4 will go through L3, L2 and L5 in sequence before arriving at L6 [38–40]. Additionally, we assume lateral cortico-cortical connections in layer 6 as they have been demonstrated in the rat barrel cortex [41] and even in layer 6b [42]. These projections are hypothesized to give rise to fast horizontal spread of single whisker-evoked excitation. To close the thalamocortical loop, the outputs of layer 6 are projected back to their somatotopically aligned barreloids and thalamic reticular population. Here we assume that neurons in the thalamic reticular nucleus (TRN) are also whisker-specific, but they are not necessarily arranged in spatially discrete clusters as barreloids.

We first focus only on the cortical network that is capable of generating SSA and true deviance detection by modulation of cross-barrel synchronous activities. All cortical intra- and inter-column synapses exhibit short-term depression. Only excitatory populations are considered in the cortical circuitry to simplify the mean-field analysis, and the level of excitability can be regulated by the threshold of neuronal gain function and the depressing effect of cortical synapses. Every layer-specific population in columns is modelled by a mean-field recurrent network coupled with depressing synapses, which can generate the so-called population spike (PS) in response to external stimulation. The PS is characterized by nearly synchronous firing of a group of neurons within a short time window [43, 44]. In the cortical mean-field rate populations we studied here, a population spike is described as a sharp increase in population activity, leading to fast depletion of mean synaptic resource that will recover gradually after the removal of external input. Higher initial synaptic resources cause more substantial PSs to be triggered in response to the same stimulus (Fig 1B).

The thalamic circuitry is set up to induce oscillations characterized by intermittent bursting activity at 7–14 Hz, which has been suggested as the potential origin of the late rhythmic responses in L4 of the anesthetized S1 [8], since the frequency of cortical oscillation ($\sim 10$Hz) falls into this range. Early studies have shown that network and intrinsic mechanisms act in combination to generate thalamocortical oscillations [45]. Two interacting populations of spiking thalamic neurons are considered to model these oscillations: excitatory

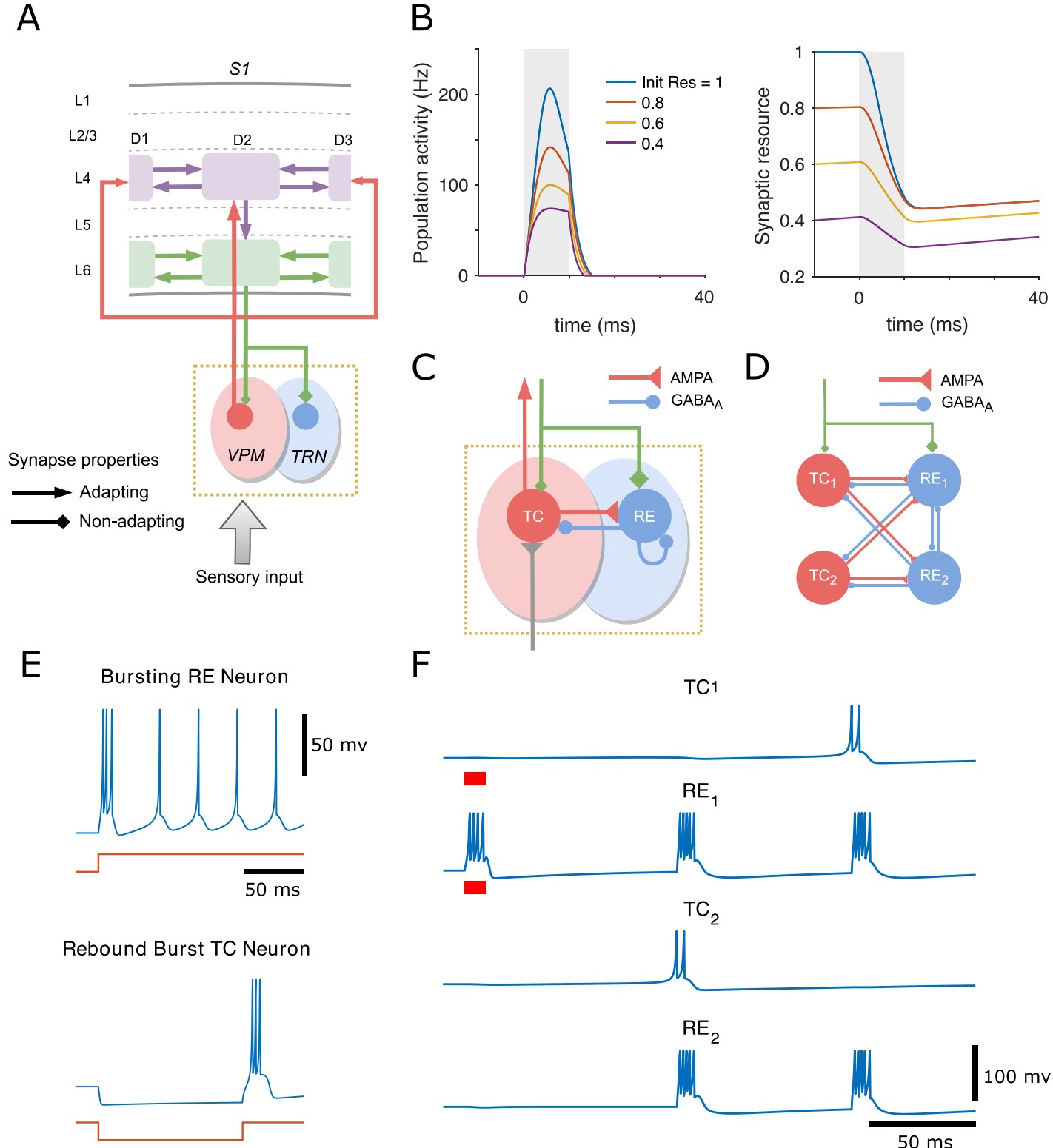

**Fig 1. Structure and dynamics of the thalamocortical network and minimal thalamic circuit inducing oscillations.** (A) The ventral posterior medial (VPM) nucleus of the thalamus (pink shade), which is somatotopically arranged into discrete clusters called "barreloids", relays peripheral sensory information to the L4 barrels (purple), with the relative strength of connections represented by the size of the arrows. Each L4 cluster projects its output vertically to the corresponding L6 infrabarrel (green), which in turn provides feedback excitation to its somatotopic barreloid and thalamic reticular nucleus (TRN, in blue shade), with higher cortical drive on TRN than VPM (specified by the size of the diamonds). Each L4 barrel and L6 infrabarrel also connects to its neighbouring barrels. (B) Population spike (PS)

can be generated in the cortical cluster, represented by the transient increase of the cluster's mean firing rate (left). Mean synaptic resources of the neuronal group are depleted by the PS and later recover gradually (right). Higher levels of initial resources evoke a more substantial PS. The time 0 represents the onset of a stimulus whose duration is marked in grey shading. (C) Two classes of cells with different synaptic receptors are used in the thalamic circuit: thalamocortical (TC) relay cells with excitatory AMPA-mediated synapses in the VPM and reticular (RE) cells with inhibitory GABA$_A$-mediated synapses in the TRN. TC and RE neuronal populations are mutually coupled, and the RE population is also recurrently connected. The thalamus can be activated by bottom-up sensory input (grey) and top-down cortical drive (green). (D) A minimal thalamic network of 2 TC and 2 RE cells driven by the cortex is capable of inducing oscillations. (E) Voltage traces of two types of thalamic cells simulated by the izhikevich model in response to a step in direct current (bottom of each panel): initial burst tonic firing in TC neuron (-70 mV at rest, hyperpolarizing pulse of 10 pA) and rebound burst in RE neuron (-62.5mV at rest, depolarizing pulse of -10 pA lasting 120 ms). (F) The time course of membrane potentials for the 4 thalamic cells. Cortical stimulation is marked by red bars under the traces (0.08 pA for TC$_1$ and 20 pA for RE$_1$).

thalamocortical (TC) neurons and inhibitory thalamic reticular (RE) neurons. As illustrated in Fig 1C, the TC population is excited by whisker stimulation and then provides AMPA-mediated excitatory synaptic input to the RE population and neurons in cortical L4. In turn, the RE neurons suppress the activity of the TC neurons and themselves via GABA$_A$-mediated inhibition. The top-down modulation from L6 exerts more effective excitation on RE than TC population due to more substantial corticothalamic conductance [36, 45, 46]. Unlike the mean-field dynamics of the cortex represented by the rate network, the intrinsic dynamics of individual spiking thalamic neuron is simulated by the izhikevich model (see Materials and methods section for model details). RE neurons exhibit transient bursting followed by tonic firing in response to depolarizing step currents, and TC relay neurons can trigger rebound spike-bursts upon release from hyperpolarization [45, 47] (Fig 1E).

We elucidate the mechanism of the oscillations triggered by the corticothalamic feedback excitation in a minimal thalamic circuit consisting of 2 RE and 2 TC cells [46] (Fig 1D). When the cortical drive is exerted on both TC$_1$ and RE$_1$ neurons, only the RE$_1$ neuron is able to elicit spikes due to the considerably higher cortical conductance on RE than TC neurons. The bursting activity of RE$_1$ cell provide inhibitory postsynaptic potentials (IPSPs) to both TC cells and then induce rebound bursting spikes in TC$_2$ cells when its potential is released from hyperpolarization and almost reaches its resting state. The firing of TC$_2$ cell reciprocally excite RE cells, leading to the next cycle of thalamocortical oscillations (Fig 1F). Minute bump in the potential of TC$_1$ cell evoked by initial corticothalamic stimulation prevents the cell from generating rebound bursts in the first cycle, which results in the TC cells fire every two cycles while RE cells fire every cycle in the minimal circuit. Through the thalamocortical pathway, these oscillations are synchronized over cortical areas as hypothesized to be the late oscillatory rhythms found in L4 of S1 [8].

## Propagation of population activities in the thalamocortical loop

The PS evoked in the principal L4 barrel will propagate to the neighbouring L4 barrels via inter-column coupling, as well as to the column-aligned L6 infrabarrel via the L4-to-L6 projection [38]. The L4 responses are largely localized around the principal barrel and substantially attenuated in the surrounding barrels [48, 49] (Fig 2A, top panel). Responses across L4 barrels start almost simultaneously about 7 milliseconds after stimulus onset, in accordance with experimental findings [50] (Fig 2B, top). The topographical spread of PS in L6 is similar to that in L4 but with broader temporal profiles of the PSs and slightly longer response latencies [50] (Fig 2A and 2B, bottom panels). The extent of propagation is determined by the width of the cross-whisker tuning curve of thalamocortical input, the strength of inter-column connections as well as the excitability of the neural population, where a broader tuning curve, stronger inter-population couplings and a more excitable population give rise to more extensive propagation.

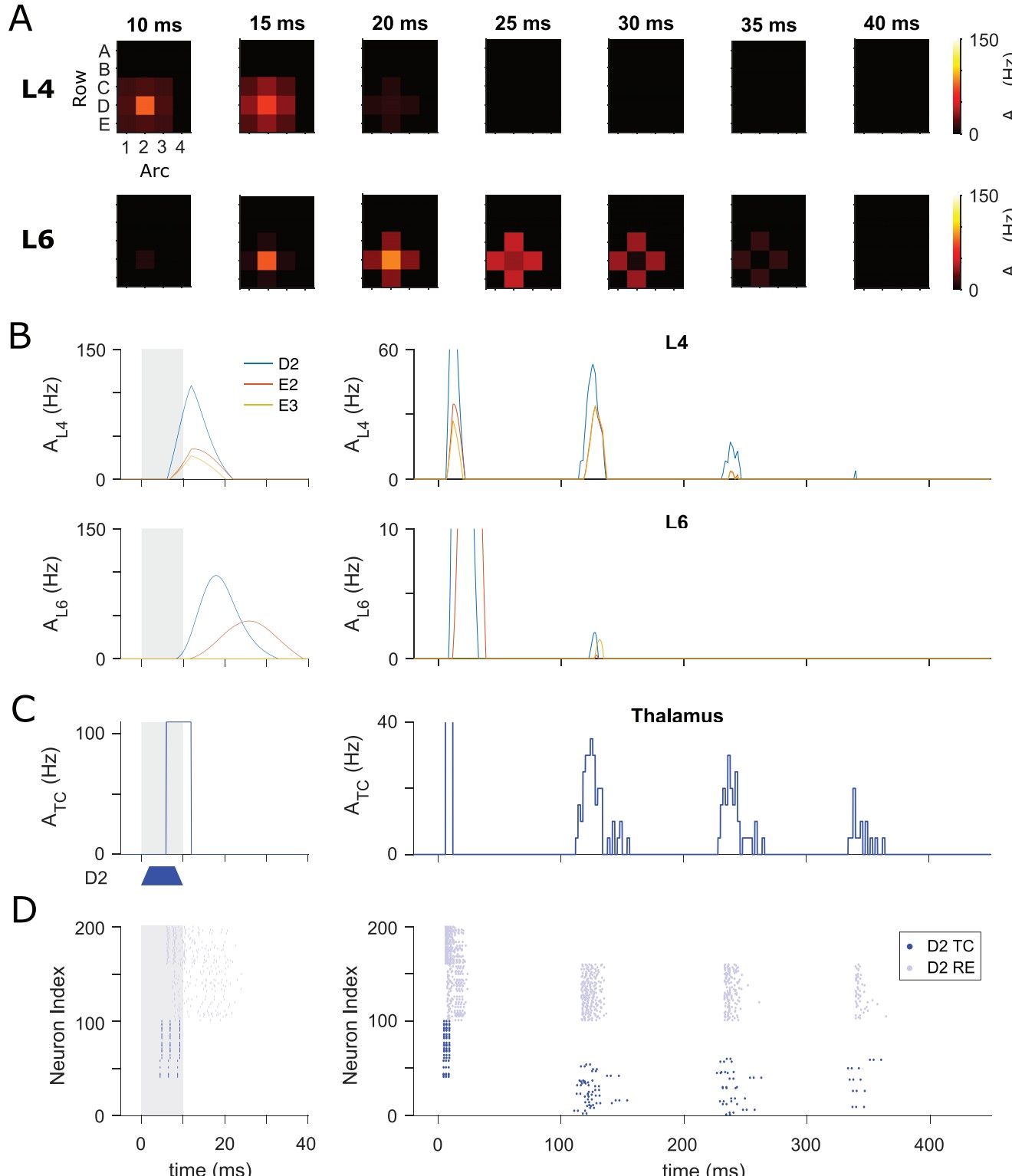

**Fig 2. Spatiotemporal distribution of whisker-evoked responses in cortical columns and thalamus.** (A) Top: Snapshots of cortical spatial activity patterns in L4, $A_{L4}$, at different times after the onset of a brief stimulation to D2 whisker. Bottom: Analogous plots for L6 populations, where population spikes occur later and over longer periods. (B) Temporal profiles of mean firing rate of three cortical columns in L4 (top) and L6 (bottom) respectively. Stimulus duration is indicated in grey shading. The timescale of the left panels matches that in (A) to show the behaviour of the initial population spikes evoked by sensory stimulation. Late spike-bursts in L4 driven by secondary thalamocortical input are displayed over a longer timescale (top-right), but the additional late

responses can hardly spread into L6 (bottom-right). (C) Population activity averaged over all 100 TC neurons in D2 barreloid, $A_{\text{TC}}$. The early synchronous activity is induced by the principal whisker deflection and relays the sensory information to the L4 barrels (left). The trapezoid form of stimulus is illustrated under the plot. The initial L6 population spike (bottom-left panel in (B)) projects back to the thalamus, eliciting the late oscillatory rhythm that begins after $\sim$110 ms of stimulation onset at an interval of $\sim$110 ms (right). (D) Spike raster for the full thalamic network of D2 barreloid (from which the population activity of TC cells shown in (C) is calculated). The circuit is composed of 100 TC cells (dark blue) in the D2 barreloid and 100 reciprocally coupled RE cells (light blue). The time axes of plots in (B), (C) and (D) are aligned.

In addition, a grid of barreloids in the VPM of the thalamus was simulated to convey sensory signals from individual whiskers to the cortex. The activity evoked by the principal whisker in barreloid D2 will not spread into other barreloids because no recurrent connections are made within the VPM, as observed experimentally [51]. The following describes the information flow within the thalamocortical system in response to a D2 whisker deflection: The D2 whisker stimulation first excites the D2 barreloid through a bottom-up connection to TC neurons, and subsequently induces transient PSs in the cortical L4 populations via thalamocortical synapses. Onset PSs are also evoked in L6 through intra-cortical connections and projected back to the thalamus, where the oscillations are initiated after the instantaneous onset response. As can be seen in Fig 2C and 2D, the TC cell group in the D2 barreloid exhibits a transient increase in population activity right after peripheral stimulation (the early response), followed by substantial synchronized activities in the 9 Hz-range (the late response). The diminishing late oscillation triggered by corticothalamic excitation occurs about 110 milliseconds after stimulus onset and lasts a couple of cycles before termination. Finally, the late thalamic activity gives rise to a secondary cortical response in L4 that only barely propagates to L6 (Fig 2B, right panels).

## SSA arises from adapted PS to standard stimulation

SSA is generally tested in the oddball protocol, where repetitive deflection is applied to one whisker (called the standard), and the sequence is randomly interrupted from time to time by deflection of another whisker (called the deviant) [4, 9]. The roles of whiskers as standard and deviant are swapped to disambiguate any response preference for individual whisker from the deflection probability on SSA.

In accordance with experimental findings [8], the L4 network exhibits prominent SSA in the late response but hardly in the initial population spike, where a similar amplitude of activity is elicited by whisker stimulation, regardless of its identity as either standard or deviant (Fig 3A, middle panel). The initial PS is relatively unaffected because the depleted synaptic resources in the L4 populations can always recover almost fully before the next stimulus is presented (Fig 3A, bottom). In contrast, depleted synaptic resources in L6 of the standard column (D2) recover only partially during the inter-stimulus interval, due to the relatively long synaptic recovery time constant in this layer (Fig 3B, blue trace in bottom panel). The insufficient resources, along with depressed L4-to-L6 input in D2 caused by repeated standard stimuli, can only trigger inadequate PSs (Fig 3B, blue trace in top panel). However, as a result of the low probability of deviant stimuli, synaptic resources in L6 of the C2 cluster often reach their steady-state (Fig 3B, red trace in bottom panel) and accordingly evoke full PSs when the deviant stimulus finally arrives (Fig 3B, red trace in top panel). In other words, SSA is initiated in L6 during the early PS phase, where the cortical L6 drive on the deviant barreloid is always strong enough to elicit thalamocortical oscillation, while it is in most cases too weak to do the same on the standard barreloid (Fig 3C and 3D). These late deviant-selective oscillations in the thalamus are then fed back to L4 populations and result in the late L4 rhythmic dynamics

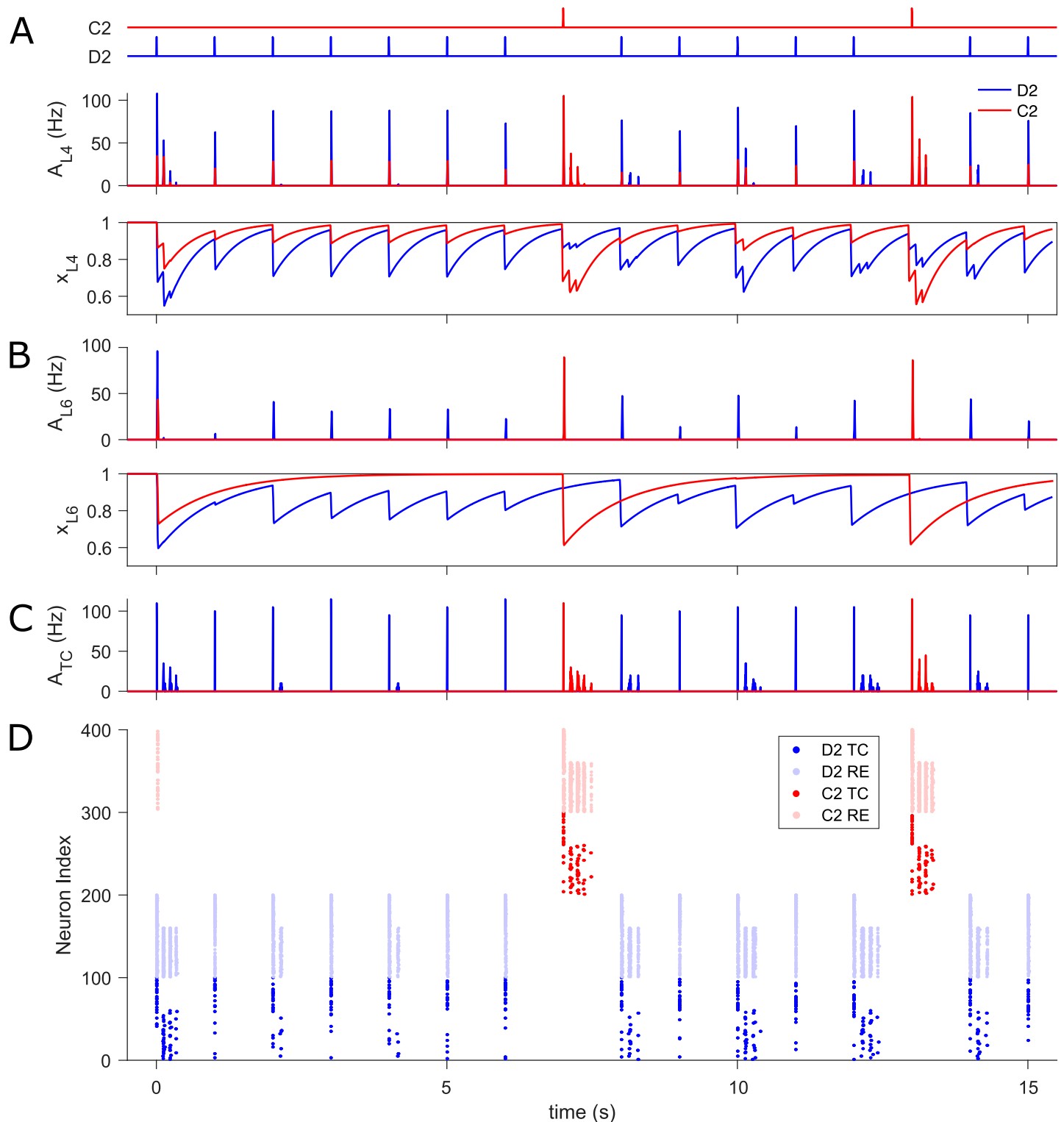

**Fig 3. SSA in the thalamocortical network.** (A) Time course of the whisker oddball protocol (top), L4 population activity, $A_{L4}$, (middle), and mean synaptic resources, $x_{L4}$, (bottom) in standard D2 (blue traces) and deviant C2 (red) barrel. Both standard and deviant stimuli evoke comparable amplitude of PS (the first burst in response to each stimulus) due to the fast recovery of depleted resources. Conversely, late oscillations occur primarily in response to deviant stimuli. (B) L6 population activity, $A_{L6}$, (top) and mean synaptic resources, $x_{L6}$, (bottom) in D2 and C2 infrabarrel. The deviant stimulus generally triggers more substantial PSs than the standard because of more resource available upon presentation of the deviant than the standard. (C) Population activities of TC cells, $A_{TC}$, in D2 and C2 barreloids. The deviant corticothalamic PS is always strong enough to initiate late thalamic oscillation, yet this is not the case for the standard. (D) Spike raster of thalamic neurons in D2 (blue)

and C2 (red) barreloids. The transient activity of C2 RE cells in response to the first stimulation are caused by the significant cortical cross-column L6 feedback. The time axes of all plots are aligned. In the oddball paradigm, peripheral stimulus duration is 10 ms and inter-stimulus interval (ISI) is 1 s (onset-to-onset).

exhibiting SSA, consistent with the experimental results that the late cortical responses are very likely to be evoked by secondary thalamocortical inputs to L4 [8].

It is worth noting that late thalamic and synchronized cortical oscillations are also sometimes elicited in D2 by the standard stimulation, and especially the standard stimulus immediately following a deviant (Fig 3C and 3D and middle panel of A), in accordance with the sporadic standard-triggered oscillation recorded in L4 of the cortex [8]. Here these late rhythmic responses occur largely because the deviant did not deplete as many resources in L6 of D2 as another standard would have done, which occasionally allows strong enough cortical drive on the thalamus to generate a thalamocortical oscillation during the presentation of the next standard stimulation.

Finally, we want to point out that our modelled thalamic response is more often not bursting than bursting and, when doing so, most prominently in response to the rare deviant stimulus, as shown in Fig 3D. On average, over trials, 25% of TC neurons in the deviant column (C2) respond in bursts to a deviant stimulus. In contrast, 10% of TC neurons in the standard column (D2) burst in response to a standard stimulus. Besides, a given bursting TC neuron in C2 (Neuron Index 210 in Fig 3D) bursts in 46% of trials of deviant stimuli, while another bursting TC neuron in D2 (Neuron Index 10) bursts in 17% of trials of standard stimuli.

## Context-dependent deviance detection

It has been suggested that the larger responses to the stimuli features, when deviant than when standard, not only reflect the rarity of the deviant causing less use-dependent adaptation [3], but also the sensitivity to the violation of the expectation established by the repeated standard, indicating a dynamic prediction mechanism for the next stimulus based on the short-term memory of the previous statistical sensory stream [52]. Such sensitivity to contextual information is termed true deviance detection/sensitivity. A common test for true deviance detection uses many-standards control protocol, where besides the standard-deviant stimuli pair in the oddball sequence, many other different stimuli are presented as the standards [19]. In this control, each of these distinct stimuli occurs with the same probability as the deviant stimulus, which consequently eliminates any potential expectation for the next stimulus but preserves the rarity of the deviant. If the response to the deviant stimulus in the oddball sequence exceeds the response to the same deviant one embedded in the control sequence, then the existence of true deviance detection is confirmed [19]. Here in the many-standards condition for the somatosensory domain, the standard stimuli are equiprobably distributed over three whiskers in a row that are adjacent to the deviant whisker, therefore each whisker is deflected with 25% probability [8] (Fig 4A).

Fig 4C illustrates the distinctive biphasic activity of C2 populations evoked by the oddball and many-standards deviant as well as of D2 populations by the oddball standard whisker deflection in the cortical L4, L6 and the thalamus. For comparison, examples of cortical multiunit (MU) recordings exhibiting late deviant-selective responses in both protocols are replotted in Fig 4B from [8]. The late responses are only found in a subset of recordings mainly confined to the L4, but with slightly different oscillatory firing patterns in individual single neurons which can result in averaging out these oscillations in the population signal. In line with the experimental findings, the simulated L4 populations elicit nearly equally strong population spikes at short latency to both the oddball standard and deviant, as well as the many-

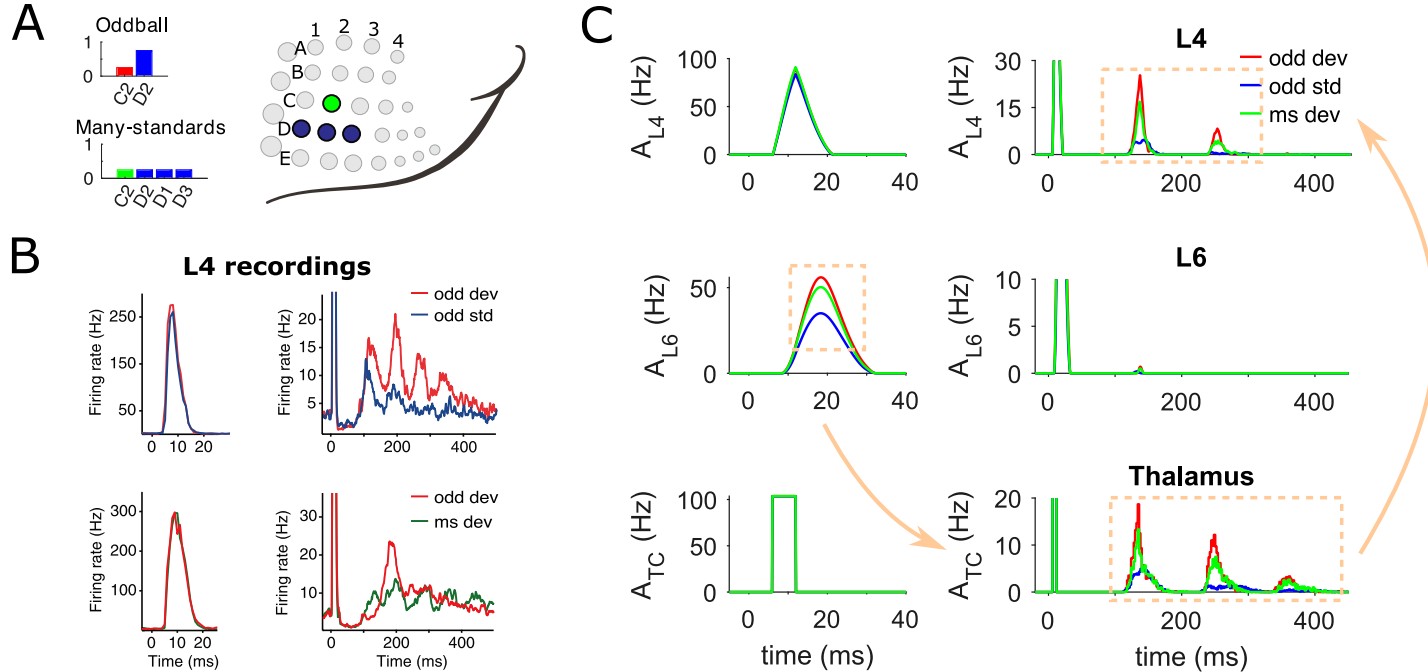

**Fig 4. Sensitivity to deviance in silico vs in vivo.** (A) Two protocols are used to test context-dependent deviance detection. In the whisker oddball condition, deviant stimuli are applied to the C2 whisker with 25% probability of occurrence, while standard stimuli are applied to the D2 whisker with 75% probability. In the many-standards condition, the deviant stimulus is presented with the same probability of appearance as in the oddball condition, but standard stimuli are equiprobably distributed over three whiskers D1-D3, each of which is stimulated 25% of the time. The bar height represents the probability of a specific whisker being stimulated. (B) A subset of multi-unit (MU) recordings in the granular layer of rat S1 shows late deviant-specific responses in the whisker oddball and many-standards protocols. Average early spike (top row, left) and late deviant-specific oscillatory responses (top row, right) are shown for standard (blue curves, activity recorded in D2) and deviant (red, activity recorded in C2) stimuli in oddball condition. The late response (bottom row) also demonstrates a distinct difference between two deviant types (both responses recorded in C2) in oddball (red curves) and many-standards protocols (green) respectively. Data are adapted from [8]. (C) Average biphasic population activity in cortical L4 (top row), L6 (middle) and thalamus (bottom) in response to oddball standard (blue traces, recorded in D2), oddball deviant (red, recorded in C2) and many-standards deviant (green, recorded in D2) stimuli. Similar to (B), the left and right columns of panels show the early and late activities respectively across cortical layers and the thalamus. SSA and true deviance detection are initiated in the early L6 responses and subsequently enhanced in the late thalamic oscillation, which finally induces the secondary deviant-selective cortical response in L4 (illustrated as flowchart with golden arrows). The third cycle of thalamic activity is too weak to produce a response in L4 since this thalamic input is below the rheobase (current threshold of gain function) of L4 neurons. Stimulation periods are marked in grey shading. The oddball and many-standards deviant responses are averaged over 30 presentations of the deviant out of 120 total stimuli in each protocol. The oddball standard responses are averaged over the complementary 90 standard stimuli. In both protocols, peripheral stimulus duration is 10 ms and inter-stimulus interval (ISI) is 1 s.

standards deviant stimulation, while SSA and true deviance detection are only observed in the late oscillatory responses (Fig 4C, top row). Importantly, our model makes further predictions of the population activity in the cortical L6 and the thalamocortical populations in both somatosensory oddball and control protocols, which, to the best of our knowledge, has not been conducted experimentally yet. We hypothesize that the late context-dependent deviant oscillation seen in L4 is reflected by the thalamic oscillation that can be solely generated by the specific thalamic circuit (Fig 4C, right panel of bottom row). The differences in the secondary thalamic responses to each type of stimulus are driven by the early cortical PS variations in the L6 circuit, which independently detects the deviant in its own responses (Fig 4C, left panel of middle row).

To understand why the average early response of the L6 network for the oddball deviant is larger than for the many-standards deviant, we need to compare the dynamics of deviant responses of L4 and L6 between the two conditions. For both the many-standards and oddball conditions, the synaptic resources of the deviant column depleted by the cross-whisker activity

of the standard(s) in L4 are able to recover to their almost full state when the subsequent stimulus is presented, due to the relatively fast recovery of L4 resources compared with the inter-stimulus interval. Therefore the L6 populations receive comparable inputs from L4 in both conditions. On the other hand, provided a slow recovery of L6 synaptic resources, the activity of the equiprobably distributed standards sometimes is strong enough to propagate into the deviant column in L6 in the many-standards condition, while the repeated oddball standard can hardly do so. Consequently, in L6, the oddball deviant column with complete resources usually evokes stronger responses than the many-standards deviant column with inadequate resources consumed by the activity of the standards, which produces the context-dependent deviance sensitivity.

Fig 5B–5D displays population activity and synaptic resources of the deviant C2 column across the cortical L4, L6, and thalamic populations, respectively. The dashed brown frames highlight two examples of an oddball deviant triggering a significant late response in L4, while a many-standards deviant does not. In the first frame, the thalamocortical inputs as the deviant in both protocols trigger approximately the same onset PS in L4 of deviant C2 (Fig 5B, top, first spike within the box). In contrast, less L6 activity is elicited in the many-standards than in the oddball condition owing to the lower level of L6 resources reduced by the last standard D3 stimulation in the many-standards condition (Fig 5C). In consequence, only the corticothalamic feedback in the oddball protocol is large enough to trigger oscillations in TC cells (Fig 5D), which in turn causes synchronized activity in the cortical L4 barrel (Fig 5B, top). In a similar vein, the second frame shows an example where resources in L6 of C2 in the many-standards condition are freshly exhausted by the preceding standard D2 stimulation and do not have time to recover adequately to trigger an oscillation. For cross-whisker adaptation in L6, resources of the deviant C2 are also occasionally depleted by the first robust burst of late rhythmic activity of standard column(s) (Fig 5C, an instance is highlighted by golden pyramid). It is worth mentioning that the two selected deviant responses in Fig 5D are only the extreme cases where late responses are almost exclusively elicited by the oddball deviant. However, overall the late thalamic responses are very variable, so the two examples shown are not particularly reflective of the mean shown in Fig 4. The many-standards deviant can induce late responses as well, although generally not as substantial as the oddball deviant. Therefore the average late responses to the oddball many-standards deviant are weaker than the oddball deviant (see S1 Fig).

Finally, we want to emphasize that the synaptic depression of intra-column connections in L6 is the primary cause of the generation of true deviance selectivity in the early L6 and late L4 responses, although depression of the thalamocortical and L4-to-L6 synapses also contributes to expanding the parameter regimes of network model giving rise to SSA.

## Further experimental predictions

We ran another pair of novel prediction paradigms on our model with the same set of parameters used in all previous protocols to test the model's sensitivity to contextual information that includes sequential statistics. Here we used variants of the many-standards paradigm with fixed deviant position but permutated the order of the standards presentations to be either random or periodic. In the sequenced paradigm, the standard deflection was periodically applied to C2, D1 and B1 whiskers, each of which in order is rarely substituted for the C1 deviant to break the regularity. Identical statistics were used in the randomized paradigm except that the three standard stimulations are randomized between the successive deviants. A schematic representation of both sequences is displayed in Fig 6A.

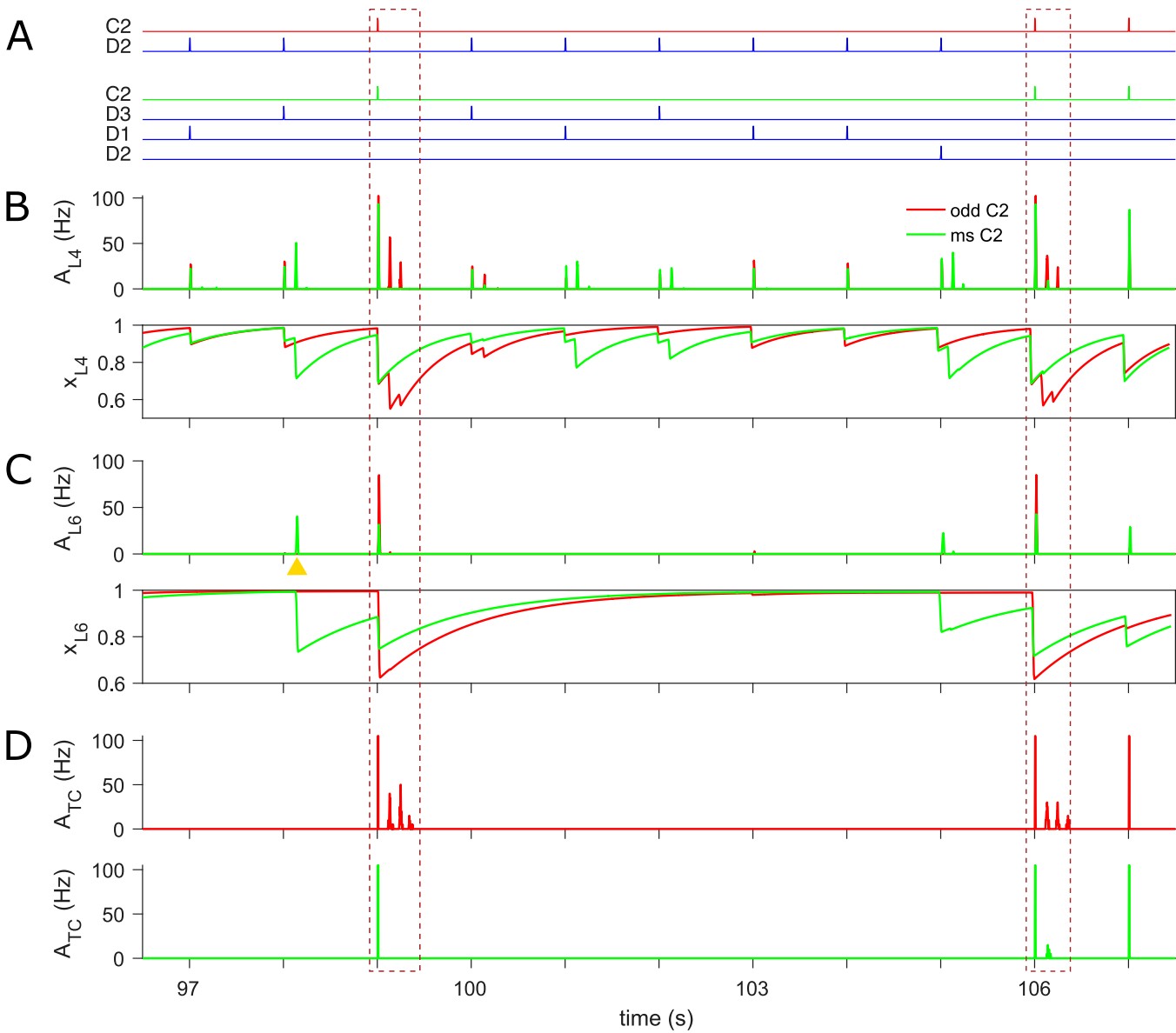

**Fig 5. True deviance detection in the model network.** (A) Illustration of oddball (top) and many-standards (bottom) protocols. The deviant stimulation to C2 is highlighted in red for the oddball condition and in green for the many-standards one. (B) Population activity (top) and synaptic resources (bottom) in L4 of C2 in response to deviant stimuli presented in oddball (red traces) and many-standards (green) condition respectively. The deviant in both protocols often evokes comparable onset responses, while a robust late oscillation is generally only induced by the oddball deviant (two typical cases are highlighted in dashed boxes). (C) Same as (B) but displayed for L6 of C2. Considerably larger early onset responses to the many-standards deviant compared to the oddball deviant are generated in the L6 population (see dashed boxes), because higher activity is propagated from the L6 standard column(s) into that of the deviant column in the many-standards compared to the oddball condition, and accordingly less synaptic resource is left and a weaker response is evoked upon arrival of the deviant. (D) Thalamocortical (TC) population activity in C2 barreloid evoked by the deviant in the oddball (red) and many-standards (green) protocols. The corticothalamic excitation of the deviant in the many-standards condition is normally strong enough to elicit late intermittent bursting, whereas that of the deviant in the oddball condition is not (dashed boxes). The time axes are aligned for panels A-D. In both stimulation paradigms, peripheral stimulus duration is 10 ms and inter-stimulus interval (ISI) is 1 s (onset-to-onset).

In comparison with the sequenced condition where the expectation is established by the repeated sequential standards and violated by the deviant, a precise prediction about forthcoming stimuli cannot be formed in the randomized condition. Our model predicts the dependence of the novelty-predicting effect on the inter-stimulus interval (ISI, onset-to-onset)

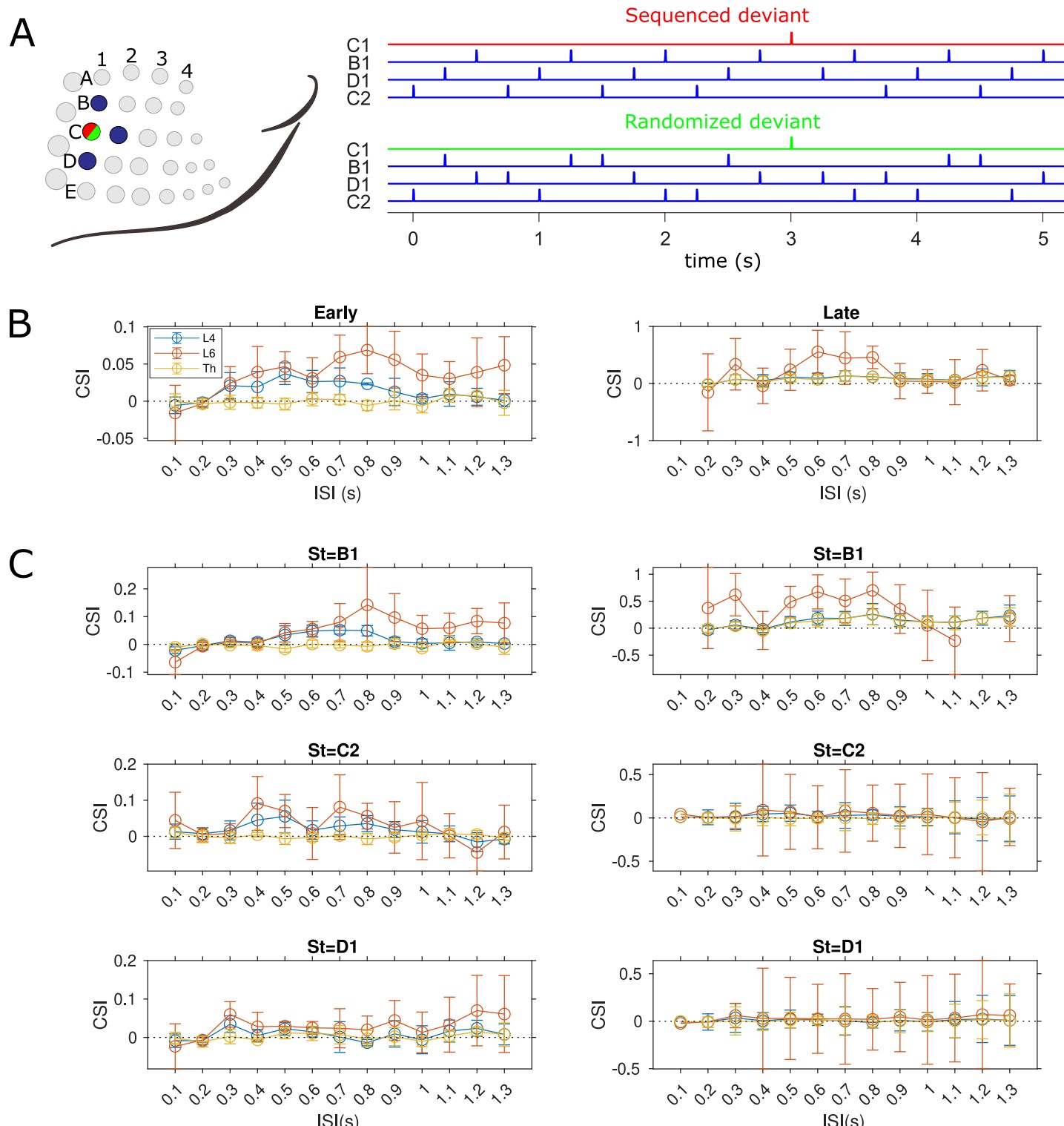

**Fig 6. Sensitivity of novelty-predicting effects to inter-stimulus intervals (ISI) tested in sequenced v.s. randomized paradigms. (A)** Illustration of sequenced and randomized stimulation trains and the stimulated whiskers. In two sequences, the delivery of the deviant stimulus to the C1 whisker is the same, but three standard stimuli are employed either periodically (sequenced condition, top) or randomly (randomized condition, bottom) to C2, D1 and B1 whiskers. The stimulus duration is 10 ms and deviant stimuli constitute 3% of overall stimuli (30 out of 1000 stimuli). **(B)** Early and late CSIs are respectively computed within time windows of 40 ms after stimulus onset and the remaining time before the onset of the next stimulus. The mean deviant responses in the CSI are averaged over 30 deviants out of 1000 stimuli in each paradigm. **(C)** Similarly, early and late CSIs are computed individually for the C1 deviant occurring right after the B1, C2 and D1 standards in the sequenced

condition. The mean responses to each deviant type are averaged over 10 deviants out of 1000 stimuli. Each condition is repeated 5 times with different random seeds to calculate the mean and variant of the CSI values.

across different cortical layers and thalamus (Fig 6B). The effect of novelty prediction is quantified by the context-specificity index (CSI), where positive values indicate the effect's existence (see the Data analysis subsection for detailed metrics descriptions). Due to the feature of biphasic dynamics in our network, CSI is separately computed for the early and late responses. For the early response, the novelty-predicting effect is demonstrated in the cortex over a wide range of ISI and particularly prominent in L6 that as we expected to be the novelty detector. Early novelty-predicting responses originated in L6 are subsequently revealed in the thalamic oscillation and secondary L4 responses via the L6-to-thalamus-to-L4 pathway.

We further investigate the individual role of three types of C1 deviants, each of which is respectively preceded by the B1, C2 and D1 standards, in the sequenced paradigm in detecting novelty across different ISIs (Fig 6C). We notice that the novelty-predicting effect is most pronounced for the type 1 deviant (in which case the deviant follows a B1 standard, St = B1), which indicates that the capacity for novelty prediction in the cortex depends on the specific sequential context in the recent past in randomized paradigms, i.e., the precise information about the order of standard presentations over hundreds of milliseconds before the arrival of the deviant.

## Discussion

We presented a cortico-thalamocortical circuit of somatosensory processing with recurrent dynamics and depressing synapses that is capable of producing both SSA and the contextual sensitivity over different latencies across cortical layers and thalamus, in line with previous experimental observations [8]. The computational results suggest that the early context-specific deviant responses in S1 L6 could be a strong candidate for the electrophysiological correlates of the mid-latency response observed in EEG recordings [21, 25], because of their similar response latencies and shared sensitivity to presentation context. Besides, we show that the late context-dependent responses in S1 L4, which temporally and functionally resemble MMN, can be produced simply by the interaction between different cortical layers and the thalamus, without involving top-down modulation by higher brain areas. It has to be noted that the previous experimental observations from [8] were recorded in anesthetized animals which may be seen as a confounding factor. However, it has been shown consistently that SSA and deviance detection of individual neurons can be observed in different sensory brain areas in both awake and anesthetized animals. For example, in the auditory system, SSA was measured in the primary auditory cortex of anesthetized of, amongst other species, cats [4], rats [53], mice [54, 55] as well as in different awake animals such as rats [17, 18] and mice [56], yielding comparable results. For the somatosensory system, SSA and deviance detection are well described in anesthetized rats [5, 57] and our own data demonstrate its relevance in awake conditions, shaping stimulus representation and perception [58]. Furthermore, MMN can be observed in different brain states, including sleeping [59], comatose [60], deeply sedated [61] or anesthetized [62] human and animal subjects [63], indicating a very robust and probably hard-wired property of central sensory processing.

Inspired by a previously described mechanism producing SSA through interactions among intracortical populations with different specificity for stimulus features in the auditory cortex [33], our model enables the generation of a cortical population spike upon the presentation of stimuli and its lateral propagation across different barrels to mediate SSA, consistent with the

given neurobiology of the barrel cortex. The existence of whisker-evoked population spikes and their spatiotemporal distribution in S1 are inferred from voltage-sensitive dye imaging [64] and multielectrode array recordings [49, 65], where the mean firing rates show a temporary peak in the principal column upon the onset of stimulation, followed by weaker activity with short latency in the surrounding columns [49]. Previous modelling work has demonstrated that this population burst could be a consequence of a recurrent network with synaptic depression [43, 44, 66, 67]. Here, such a network is the essential ingredient of our hierarchical model to produce SSA.

Our model can be regarded as a simplified representation of the cortico-thalamocortical pathway of the rodent whisker system, concentrating on the minimal number of components necessary to give rise to deviant signals. Due to this "minimalistic" approach, we left out other parts of the cortical circuitry, such as layers 2/3 and 5, which are arguably involved in processing deviant stimuli and probably key to forwarding these signals further downstream of the cortical hierarchy [8, 53]. The dynamics of the barrel cortex are described by population activity and mean synaptic resources of somatotopic neuronal clusters in a two-layered grid network. The reasons for adopting rate models for the cortical dynmaics are twofold. First, the mean-field description of population activity makes it possible to analyze the dynamics of single recurrent clusters giving rise to population spikes on the phase plane, under the mathematical abstraction that each cortical cluster is a large and homogenous network. Theoretical work has demonstrated that such a mean-field representation is equivalent to a homogeneous pool of leaky integrate-and-fire neurons in terms of its population activities [68]. Second, this specification allows us to focus on the collective population behaviour within the grid network, which is sufficient to generate cortical SSA and true deviance detection. SSA and true deviance sensitivity cannot be accounted for solely by the intrinsic dynamics of individual neurons, but relies to a large extent on network dynamics. It should be noted that although this mean-field network of the barrel cortex ignores some details, such as the potential contribution of additional layers, distinct neuron types and their morphology, and more realistic connectivity profiles, critical features of the barrel cortex including its somatotopic columnar arrangement and the spatiotemporal pattern of whisker-evoked responses are retained and indeed shown to be sufficient to explain the deviant selectivity of transient L6 responses.

For the thalamic circuit, we utilized an integrate-and-fire model (such as the Izhikevich model) with synaptic kinetics of AMPA and GABA receptors to generate thalamic bursting spiking dynamics, which reproduces, with higher computational efficiency, the spiking patterns of spindles modelled by a more detailed Hodgkin–Huxley model proposed by Destexhe et al. [46]. Here, our goal was not to model actual sleep spindles but to build a phenomenological network model that reproduces thalamic rhythmic activity. This strategy to utilize Destexhe's model [46], or at least some of its parameters, has been used before to model other phenomena with thalamic involvement, such as pathogenesis in childhood absence epilepsy [69], or more generally to model "[. . .] *rhythmogenesis, involving post-inhibitory rebound excitation in the TC neurons*" [70]. We also want to point out that SSA paradigms generally use carefully controlled passive stimulation rather than active touch (e.g., detecting an object in the dark). Many studies have reported bursting activity in the thalamus similar to our model when whisker stimuli were applied passively both in anesthetized [71, 72] and awake animals [71, 73], and spontaneously in awake animals as well [74, 75]. While post-inhibitory rebound bursting activity of thalamic neurons is characteristic of the unconscious state, it can be observed in the auditory thalamus of awake, passively listening animals as well in the form of a post-stimulus burst [76], even though at longer stimulus durations than used in the model presented here. In contrast to the cortical rate model, which only depends on network mechanisms, the thalamic spiking model takes into account both the intrinsic properties of single

cells and their interplay to elicit synchronous oscillations. Our modelling results implicate the slow recovery variable in these thalamocortical neurons, which may describe the inactivation of low-threshold $Ca^{2+}$ T-currents [77], as the main factor driving the latency of the secondary response observed in S1 L4 and therefore potentially MMN.

Finally, it is worth mentioning that the current implementation of our model still has several shortcomings. One limitation is the sensitivity of SSA to cortical network parameters, which stems from the different layer-specific response patterns in the feedforward L4-to-L6 network. In other words, the parameter regime allowing for SSA and the true deviance sensitivity in initial L6 responses is constrained by not only the spatiotemporal pattern of whisker-evoked transient activity in the L6 network but also responses in L4 that directly delivers its output to L6. Primarily, in the L6 circuit, the PS evoked by the oddball standard should reduce with repeated stimuli while not generalize to the oddball deviant. Besides, the propagation of activity across the L6 network is supposed to be robust enough to occasionally pass the PSs triggered by standards in the many-standards condition into the deviant column, yet too weak for PSs evoked by the oddball standard to do so. Another possible shortcoming of our minimalistic model is that it omitted cortical layer 2/3 and layer 5 that could modulate the laminar pathway from L4 to L6. Future work incorporating these missing layers could generate richer dynamics that expand the narrow region of SSA existence in the parameter space, leading to a robust SSA phenomenon. Lastly, it would be instructive to test the NMDA dependency of the late response in a spiking implementation of the cortical dynamics explored here to further investigate SSA's connection with MMN.

## Materials and methods

### Derivation of spiking to rate network

We derive the rate equations for a network of interacting populations mediated by depressing synapses assuming that each population is large and homogeneous, using standard techniques [68, 78–80]. By homogenous, we mean that (i) all neurons have the same properties; (ii) each neuron receives identical external input; and (iii) the coupling strength between any pair of neurons within the population is statistically uniform. The general conclusions developed here will be applied in the next section to formalize the mean-field cortical network composed of a grid arrangement of coupled populations.

We define the mean firing rate as the population activity $A(t)$ averaged over a pool of homogenous neurons.

$$A(t) \equiv \lim_{\Delta t \to 0} \frac{n_{\text{act}}(t; t + \Delta t)}{N \Delta t} = \frac{1}{N} \sum_{j=1}^{N} \sum_{f} \delta(t - t_j^f) \qquad (1)$$

where $N$ is the number of neurons within the population, $n_{\text{act}}(t; t + \Delta t)$ denotes the number of spikes of all neurons in the population, which appear betweem time $t$ and $t + \Delta t$. Since $\Delta t$ is a very short time interval, each neuron fires at most one spike during the interval and correspondingly $n_{\text{act}}(t; t + \Delta t)$ also represents the number of neurons that fire between time $t$ and $t + \Delta t$. $\delta(t)$ is the Dirac delta function and $t_j^f$ denotes the firing time of the $j$-th neuron in the population. This notion of the rate as population activity averaged over many neurons will be widely used in the following derivation.

The dynamics of short-term depression for the connection between pre- and post-synaptic neuron pair $j$, $i$ is given by the phenomenological model [66, 81]:

$$\frac{\mathrm{d}x_{ij}(t)}{\mathrm{d}t} = \frac{1 - x_{ij}(t)}{\tau_{\mathrm{rec}}} - Ux_{ij}(t)\sum_f \delta(t - t_j^f) \tag{2}$$

where each pre-synaptic spike emitted at $t_j^f$ utilizes a certain rate $U$ of the available synaptic resource $x_{ij}$ that is a fraction of full resources. The depleted available resource then returns to its baseline value with a time constant $\tau_{\mathrm{rec}}$. For the current-based type of post-synaptic input, which is independent of neuronal membrane potential, the increment in the amplitude of post-synaptic current $I_i(t)$ triggered by a pre-synaptic spike arriving at $t_j^f$ is given by

$$\Delta I_i(t_j^f) = JUx_{ij}^-(t_j^f) \tag{3}$$

where $J$ is the absolute synaptic efficacy and $x_{ij}^-(t_j^f)$ denotes the value of $x_{ij}$ immediately before the arrival of the spike.

We define $x_i(t) \equiv 1/N \sum_{j=1}^N x_{ij}$ as the mean available resource in a bunch of synapses from a population of pre-synaptic neurons $j$ to a post-synaptic neuron $i$. By taking the average over the pool of pre-synaptic neurons on both sides of Eq 2, we get

$$\frac{\mathrm{d}x_i(t)}{\mathrm{d}t} = \frac{1 - x_i(t)}{\tau_{\mathrm{rec}}} - \frac{1}{N}\sum_{j=1}^N Ux_{ij}(t)\sum_f \delta(t - t_j^f) \tag{4}$$

The second term on the right-hand side of Eq 4 can be further transformed, assuming uncorrelated Poisson spiking from a large population of pre-synaptic neurons, to

$$\frac{U}{N}\sum_{j=1}^N\sum_f x_{ij}(t)\delta(t - t_j^f) \quad = U\lim_{\Delta t \to 0}\frac{\sum_{j \in \Omega}x_{ij}(t)}{N\Delta t}$$

$$\approx U\lim_{\Delta t \to 0}\frac{x_i(t)n_{\mathrm{act}}(t; t + \Delta t)}{N\Delta t} \quad = Ux_i(t)A(t) \tag{5}$$

where $N$ is the number of neurons in the pre-synaptic population. $\Omega$ denotes the subpopulation of pre-synaptic neurons that fire between time $t$ and $t + \Delta t$, and $n_{\mathrm{act}}(t; t + \Delta t)$ is the size of the subpopulation. The approximation becomes an equality in the limit of $N \to \infty$. $A(t)$ represents the population activity of pre-synaptic neurons as defined in Eq 1.

Using Eqs 5 and 4 can be reformulated as

$$\frac{\mathrm{d}x_i(t)}{\mathrm{d}t} = \frac{1 - x_i(t)}{\tau_{\mathrm{rec}}} - Ux_i(t)A(t) \tag{6}$$

We find that the dynamics of the mean synaptic resource $x_i(t)$ is independent of the neuronal identity and therefore is the same for all neurons in the population.

In addition, we assume that each pre-synaptic spike emitted at $t_j^f$ triggers an instantaneous post-synaptic current with the temporal profile $\alpha(t - t_j^f)$ and the efficacy for each synaptic connection in an all-to-all coupled population of $N$ neurons is scaled as $J_0/N$. The input current to a neuron $i$ in the population is generated by all spikes of all neurons within the same

population,

$$I_i(t) = \sum_{j=1}^{N}\sum_f \frac{J_0}{N} U x_{ij}^-(t_j^f)\alpha(t - t_j^f)$$

$$= J_0 U \sum_{j=1}^{N}\sum_f \int_0^\infty \frac{1}{N} x_{ij}(t - s)\alpha(s)\delta(t - t_j^f - s)\mathrm{d}s \tag{7}$$

$$= J_0 U \int_0^\infty \alpha(s)\frac{1}{N}\sum_{j=1}^{N}\sum_f x_{ij}(t - s)\delta(t - t_j^f - s)\mathrm{d}s$$

Making use of Eq 5 with $t$ replaced by $t - s$ to substitute the average quantity on the right-hand side of Eq 7, we obtain

$$I(t) = J_0 U \int_0^\infty \alpha(s)x(t - s)A(t - s)\mathrm{d}s \tag{8}$$

We dropped the neuronal index $i$ since the input current at time $t$ depends on past mean synaptic resource and population activity and is identical for all neurons.

Now we generalize the arguments from a single fully-connected pool to multiple interacting pools. Under the assumption that neurons are homogenous in each population, the activity of neurons in population $k$ is

$$A_k(t) = \frac{1}{N_k}\sum_{j\in\Gamma_k}\sum_f \delta(t - t_j^f) \tag{9}$$

where $N_k$ is the size of population $k$ and $\Gamma_k$ represents the set of neurons that belongs to population $k$.

We assume that each neuron $i$ in population $k$ receives input from all neurons $j$ in population $l$ with adapting coupling strength $(J_{kl}/N_l)U_{kl}x_{ij}(t)$ and the post-synaptic current of the neuron $i$ elicted by a spike of a presynaptic neuron $j$ has the time course $\alpha_{kl}(t)$. Here the synaptic efficacy $J_{kl}$, utilization rate $U_{kl}$ and post-synaptic current $\alpha_{kl}(t)$ depend on the type of synaptic connection from a neuron in population $k$ to a neuron in population $l$ rather than the neuronal identity. The input current to a neuron $i$ in pool $k$ is induced by all spikes of all neurons in the network of pools,

$$I_{i,k}(t) = \sum_l \sum_{j\in\Gamma_l}\sum_f \frac{J_{kl}}{N_l} U_{kl} x_{ij}^-(t_j^f)\alpha_{kl}(t - t_j^f)$$

$$= \sum_l J_{kl} U_{kl} \int_0^\infty \alpha_{kl}(s)\frac{1}{N_l}\sum_{j\in\Gamma}\sum_f x_{ij}(t - s)\delta(t - t_j^f - s)\mathrm{d}s \tag{10}$$

$$= \sum_l J_{kl} U_{kl} \int_0^\infty \alpha_{kl}(s)x_i(t - s)A_l(t - s)\mathrm{d}s$$

Correspondingly the dynamics of mean synaptic resource $x_i(t)$ is generalized as

$$\frac{\mathrm{d}x_i(t)}{\mathrm{d}t} = \frac{1 - x_i(t)}{\tau_{\mathrm{rec},kl}} - U_{kl}x_i(t)A_l(t) \tag{11}$$

By assuming that $U_{kl}$ and $\tau_{\text{rec},kl}$ are homogenous across different pools, we drop the subscript $kl$ and reformulate the above equation as

$$\frac{\mathrm{d}x_l(t)}{\mathrm{d}t} = \frac{1 - x_l(t)}{\tau_{\text{rec}}} - Ux_l(t)A_l(t) \tag{12}$$

Here we change the index of mean synaptic resource from post-synaptic neuron $i$ in population $k$ to pre-synaptic population $l$ since the evolution of mean synaptic resource is governed by the activity of pre-synaptic population and is the same for all neurons within post-synaptic populations.

In this case, the input current,

$$I_k(t) = \sum_l J_{kl} U \int_0^\infty \alpha_{kl}(s) x_l(t - s) A_l(t - s) \mathrm{d}s \tag{13}$$

which is independent of the neuronal index $i$ but the post-synaptic population index $k$.

Finally, we characterize the dynamics of the input current $h_k(t)$ of population $k$ with the differential equation of passive membrane and employ for each population the rate model $A_l(t) = F_l(h_l(t))$ in which $F_l$ is the stationary gain function of neurons in population $l$. The input current $h_k(t)$ takes into account both synaptic coupling $I_k(t)$ and external drive $I_k^{\text{ext}}(t)$,

$$\tau_{\text{m}} \frac{\mathrm{d}h_k(t)}{\mathrm{d}t} = -h_k(t) + \sum_l J_{kl} U \int_0^\infty \alpha_{kl}(s) x_l(t - s) F_l(h_l(t - s)) \mathrm{d}s + I_k^{\text{ext}}(t) \tag{14}$$

where all neurons in the network have the same membrane time constant $\tau_{\text{m}}$.

Particularly, in the case of instantaneous post-synaptic current pulse $\alpha_{kn}(s) = \delta(s)$, Eq 14 can be reduced to a first-order differential equation

$$\tau_{\text{m}} \frac{\mathrm{d}h_k(t)}{\mathrm{d}t} = -h_k(t) + \sum_l J_{kl} U x_l(t) F_l(h_l(t)) + I_k^{\text{ext}}(t) \tag{15}$$

## Modelling the thalamocortical network of the barrel cortex

It has been observed that a cortico-thalamocortical loop structure exists in the rodent whisker system [37]. The excitatory thalamocortical (TC) neurons in the ventral posterior medial (VPM) nucleus present feedforward projection to layer 4 (L4) of the cortex. On the other hand, neurons in layer 6 (L6) provide feedback excitation to TC and reticular(RE) neurons that impose reciprocal inhibition on TC cells.

**Model of the cortical part.** Here, for the complexity needed for the work, we greatly simplified the laminar structure of the cortex by only considering two cascaded excitatory networks that model L4 and L6 respectively. To mimic the somatotopic organization of the barrel cortex, the circuit of each layer is composed of an array of cortical columns (barrels). Each column is modelled by a fully connected excitatory population mediated by synaptic depression, with its mean-field dynamics as analyzed in the last subsection. Every column is also connected to its nearest vertical and horizontal neighbours by inter-column depressing synapses. Furthermore, the L4 columns receive thalamic inputs with cross-whisker tuning and send outputs to their aligned L6 columns.

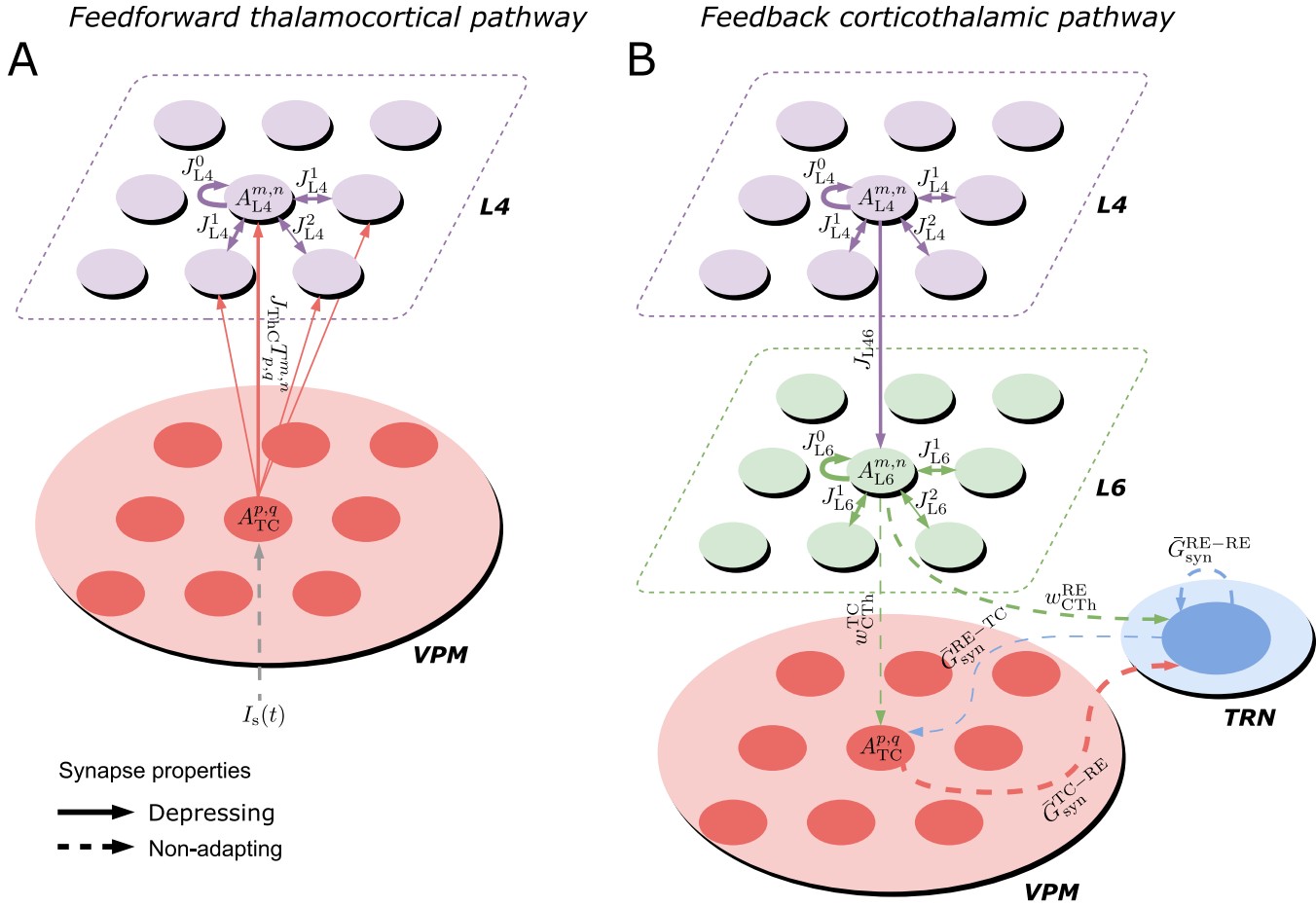

**Fig 7. Schematic representation of the thalamo-cortio-thalamic loop model of the barrel cortex, to illustrate the notation used in Methods.** (A) Feedforward thalamocortical pathway. (B) Feedback corticothalamic pathway. The width of arrows describes the relative strength of each connection.

Based on Eqs 15 and 12, the mean-field dynamics of cortical layer 4 network is described by the following equations (see Fig 7 for a schematic representation of the network architecture):

$$\tau_m \frac{dh_{L4}^{m,n}}{dt} = -h_{L4}^{m,n} + \sum_{k=-1}^{1}\sum_{l=-1}^{1} J_{L4}^{|k|+|l|} U_{L4} x_{L4}^{m+k,n+l} A_{L4}^{m+k,n+l}$$
$$+ \sum_{p=1}^{M}\sum_{q=1}^{N} J_{ThC} T_{p,q}^{m,n} U_{ThC} z_{p,q}^{m,n} A_{TC}^{p,q} \tag{16}$$

$$\frac{dx_{L4}^{m,n}}{dt} = \frac{1 - x_{L4}^{m,n}}{\tau_{rec,L4}} - U_{L4} x_{L4}^{m,n} A_{L4}^{m,n} \tag{17}$$

$$\frac{dz_{p,q}^{m,n}}{dt} = \frac{1 - z_{p,q}^{m,n}}{\tau_{rec,ThC}} - T_{p,q}^{m,n} U_{ThC} z_{p,q}^{m,n} A_{TC}^{p,q} \tag{18}$$

$$A_{L4}^{m,n} = \left[\alpha_{L4}(h_{L4}^{m,n} - \theta_{L4})\right]_+ \tag{19}$$

Similarly, the mean-field cortical layer 6 network is defined as:

$$\tau_m \frac{\mathrm{d}h_{\mathrm{L6}}^{m,n}}{\mathrm{d}t} = -h_{\mathrm{L6}}^{m,n} + \sum_{k=-1}^{1}\sum_{l=-1}^{1} J_{\mathrm{L6}}^{|k|+|l|} U_{\mathrm{L6}} x_{\mathrm{L6}}^{m+k,n+l} A_{\mathrm{L6}}^{m+k,n+l} + J_{\mathrm{L46}} U_{\mathrm{L46}} x_{\mathrm{L46}}^{m,n} A_{\mathrm{L4}}^{m,n} \tag{20}$$

$$\frac{\mathrm{d}x_{\mathrm{L6}}^{m,n}}{\mathrm{d}t} = \frac{1 - x_{\mathrm{L6}}^{m,n}}{\tau_{\mathrm{rec,L6}}} - U_{\mathrm{L6}} x_{\mathrm{L6}}^{m,n} A_{\mathrm{L6}}^{m,n} \tag{21}$$

$$\frac{\mathrm{d}x_{\mathrm{L46}}^{m,n}}{\mathrm{d}t} = \frac{1 - x_{\mathrm{L46}}^{m,n}}{\tau_{\mathrm{rec,L46}}} - U_{\mathrm{L46}} x_{\mathrm{L46}}^{m,n} A_{\mathrm{L4}}^{m,n} \tag{22}$$

$$A_{\mathrm{L6}}^{m,n} = [\alpha_{\mathrm{L6}}(h_{\mathrm{L6}}^{m,n} - \theta_{\mathrm{L6}})]_+ \tag{23}$$

The dynamics of each barrel is described by its population activity $A^{m,n}$ and mean fraction of synaptic resource available $x^{m,n}$, where $m$ and $n$ denote the row and arc indices of the barrel on an idealized $M \times N$ grid. Both variables are specified by subscript either L4 for the population in layer 4 or by L6 for layer 6 (which applies for all layer-specific variables). The population activity $A^{m,n}$ is defined as a threshold- linear gain function of the mean-field current $h^{m,n}$ ($\alpha$ and $\theta$ are respectively the slope and horizontal intercept of the gain function). The function $[\cdot]_+$ is defined as its argument when the argument is positive, otherwise zero.

The synaptic efficacy is represented by $J^{|k|+|l|}$ with superscript specifying the column-to-column distance of the connection (0, 1 and 2 respectively indicate intra-column, vertical/horizontal and diagonal inter-column connection). $U$ denotes the utilization rate of synaptic resources. $\tau_m$ and $\tau_{rec}$ are the membrane time constant of cortical neurons and recovery time constant of synaptic resources, repectively. The dynamics of thalamocortical synapses from the whisker channel $p, q$ to the L4 barrel $m,n$ is characterized by the fraction of resources available $z_{p,q}^{m,n}$, parameterized by utilization rate $U_{\mathrm{ThC}}$ and recovery time constant $\tau_{\mathrm{rec,ThC}}$. Similarly, the L4-to-L6 depressing synapse of column $m,n$ is described by its resources $x_{\mathrm{L46}}^{m,n}$ with utilization rate $U_{\mathrm{L46}}$ and recovery time constant $\tau_{\mathrm{rec,L46}}$. $J_{\mathrm{ThC}}$ and $J_{\mathrm{L46}}$ denote the efficacy of thalamocortical and laminar synapses, respectively.

$T_{p,q}^{m,n}$ represents the relative magnitude at which the barrel $m,n$ receives thalamocotical input from whisker channel $p, q$ compared with its primary whisker channel. The values of $T_{p,q}^{m,n}$ over all channels on the $M \times N$ grid compose the spatial tuning curve of the thalamocortical inputs to the barrel $m,n$. We chose a linear tuning curve in the simulation, which is defined as

$$T_{p,q}^{m,n} = [-d/\lambda + 1]_+ \tag{24}$$

where $\lambda$ is the radius of tuning curve and $d$ is the floored Euclidian distance between barrel $m, n$ and whisker channel $p, q$ on the evenly-spaced gird where the separation distance is 1

$$d = \lfloor \sqrt{(m-p)^2 + (n-q)^2} \rfloor \tag{25}$$

where $\lfloor \cdot \rfloor$ denotes the floor function that rounds its argument to the nearest integer less than or equal to that argument.

$A_{\mathrm{TC}}^{p,q}$ describes the thalamocortical input delivered through the whisker channel $p, q$. It is defined as the population activity of TC cells organized in barreloid $m,n$, which in practice is evaluated by Eq 1 with a time window $\Delta t = 2$ ms. It is worth noting that the time bin of 2 ms is only used to compute the population activity of TC cells from their spikes, and the cortical population activity is obtained by numerically solving those differential equations of the

cortical dynamics with an integration time step of 0.1 ms. Specifically, the $A_{TC}^{p,q}$ is updated every time window (2 ms) and held constant for 20 integration time steps (each 0.1 ms).

**Model of the thalamic part.** We assume that the late cortical rhythmic activity (roughly 10 Hz) possibly emerges from the thalamocortical system that can present thalamocortical oscillatory activity with a frequency range of 7–14 Hz [45, 46]. Motivated by the anatomy of the VPM and reticular nucleus of the thalamus [82], we modelled each barreloid in the VPM nucleus by a cluster of 100 TC cells mediated by a pool of 100 RE cells, whose connectivity is illustrated in Fig 7B. The spiking dynamics of each thalamic neuron simulated by the Izhikevich model, which uses the quadratic integrated-and-fire model for the membrane potential equation, is expressed as

$$\dot{v} = 0.04v^2 + 5v + 140 - u + I(t)$$
$$\dot{u} = a(bv - u)$$

(26)

with after-spike resetting

$$\text{if } v \geq +30\text{mV}, \quad \text{then } \begin{cases} v \leftarrow c \\ u \leftarrow u + d \end{cases}$$

(27)

where the time $t$ is in milliseconds; $v$ and $u$ respectively denote the membrane potential (in mV) and recovery variable that represents the difference of all influx and efflux of voltage-gated ionic currents. In particular, the population of 100 TC cells in each barreloid are divided into two subgroups. $(a, b, c, d) = (0.005, 0.26, -52, 2)$ are assigned to the first 60 TC neurons (indexed from 1 to 60) to induce rebound bursts that is an essential ingredient of the thalamo-cortical oscillation. The same parameter quartet except for $b = 0.25$ is assigned to the rest 40 TC neurons (indexed from 60 to 100) that only fire spike trains following stimulation from depolarized currents but fail to produce rebound spikes. Thalamic RE cells are modelled as bursting neurons by assigning $(a, b, c, d) = (0.02, 0.2, -55, 4)$. Finally, it is worth mentioning that the reason that we chose the proportion of post-inhibitory bursting TC neurons to all TC neurons $P_{TC} = 0.6$ is to model the decaying thalamic oscillation by trading off the duration and strength of the thalamic rhythmic bursting evoked by the initial cortical feedback. A more substantial but less sustained rhythmic firing pattern is generated after the first transient response as the rate of post-inhibitory bursting TC cells $P_{TC}$ increases (see S2 Fig).

The network connectivity is illustrated in Fig 1C, with a connection probability of 0.6 for TC-to-RE, RE-to-TC and RE-to-RE coupling in the two pairs of equal-sized populations of TC and RE cells and 0.2 for coupling between two subgroups of RE cells. $I(t)$ is the total current input to a thalamic cell (in pA)

$$I(t) = I_{noise}(t) - I_{syn}(t) + I_{CTh}(t) + I_s(t)$$

(28)

The thalamic background noise $I_{noise}(t)$ is subject to uniform distribution bound between -0.5 and 0. The conductance-based synaptic current is modelled as

$$I_{syn}(t) = g_{syn}(t)(v(t) - E_{syn})$$

(29)

where $E_{syn}$ is the synaptic reversal potential. Synaptic conductivity $g_{syn}(t)$ has a time course of exponential decay:

$$g_{syn}(t) = \sum_f \bar{g}_{syn} e^{-(t-t^f)/\tau} \Theta(t - t^f)$$

(30)

where $\Theta(t)$ is the Heaviside step function and $t^f$ represent the arrival time of pre-synaptic spikes. We simulated AMPA and GABA$_A$ receptors with the decay time constant $\tau = 5$ and 6 ms; $E_{syn} = 0$ and -75 mV, respectively. The maximal conductance $\bar{g}_{\text{syn}}$ for each type of connection is scaled as $\bar{g}_{\text{syn}} = \bar{G}_{\text{syn}}/C$, where $C$ is the number of randomly chosen presynaptic partners for each neuron and = 2, 0.01 and 0.5 nS for AMPA-mediated TC-to-RE, GABA$_A$-mediated RE-to-TC and GABA$_A$-mediated RE-to-RE synapses, respectively.

In addition, half of TC cells in each barreloid and their coupled RE cells were randomly picked receiving corticothalamic feedback current, which is expressed as

$$I_{\text{CTh}}(t) = w_{\text{CTh}}A_{\text{L6}}(t) \tag{31}$$

where $A_{\text{L6}}(t)$ denotes the population activity of L6 neurons in the somatotopic infrabarrel. We respectively assigned the corticothalamic coupling strength $w_{\text{CTh}} = 0.001$ and 0.4 for TC and RE neurons.

Finally, we respectively stimulated randomly 10% and 50% of TC cells in the two complementary subgroups of each barreloid with sensory input that is described as

$$I_{\text{s}}(t) = \xi(t) \cdot B \tag{32}$$

where $B$ and $\xi_{p,q}(t)$ respectively represent the maximum amplitude and temporal envelope of the sensory stimulus (normalized between 0 and 1). In all simulation protocols, $B = 5$ and $\xi_{p,q}(t)$ has the profile of trapezoid pulse with 10 ms duration (2 ms onset/offset ramp time).

Forward Euler method with a time step of 0.1 ms is used to simulate the network dynamics. The values for different network parameters are listed in Table 1.

## Geometrical analysis of the thalamic rhythmic bursting

It is worth mentioning that the parameters of thalamic neurons and their connectivity strength need to be finely tuned to demonstrate the decaying rhythmic bursting behavior shown in Fig 2D. In order to understand the mechanisms that mediate the termination of the rhythmic activity in our model, we first illustrate in the phase portrait of the Izhikevich model how the post-inhibitory bursts of the TC cells can be triggered by the activation of RE cells (Fig 8). A

**Table 1. Values used for the network parameters.**

| Notation | Description | Value |
|---|---|---|
| $M$ | Number of rows | 5 |
| $N$ | Number of arcs | 4 |
| $\lambda$ | Radius of tuning curve | 1.6 |
| $U_{\text{L4}}/U_{\text{L6}}$ | Utilization rate of L4 / L6 synapses | 0.5 |
| $U_{\text{ThC}}/U_{\text{L46}}$ | Utilization rate of thalamocortical / L4-to-L6 synapses | 0.8 / 0.5 |
| $\alpha_{\text{L4}}/\alpha_{\text{L6}}$ | Slope of L4 / L6 gain function | 1 |
| $\theta_{\text{L4}}/\theta_{\text{L6}}$ | Threshold of L4 / L6 gain function | 5 / 3 |
| $J_{\text{L4}}^0/J_{\text{L6}}^0$ | Intra-column synaptic efficacy in L4 / L6 | 2.2 / 2.5 |
| $J_{\text{L4}}^1/J_{\text{L6}}^1$ | Vertical/horizontal inter-column synaptic efficacy in L4 / L6 | 0.05 / 0.03 |
| $J_{\text{L4}}^2/J_{\text{L6}}^2$ | Diagonal inter-column synaptic efficacy in L4 / L6 | 0.001 |
| $J_{\text{ThC}}$ | Synaptic efficacy of thalamocortical connection | 1 / 0.05 |
| $J_{\text{L46}}$ | Synaptic efficacy of L4-to-L6 connection | 0.24 |
| $\tau_m$ | Membrane time constant | 0.001 s |
| $\tau_{\text{rec,L4}}/\tau_{\text{rec,L6}}$ | Recovery time constant of L4 / L6 synaptic resources | 0.5 / 1 s |
| $\tau_{\text{rec,ThC}}/\tau_{\text{rec,L46}}$ | Recovery time constant of thalamocortical / L4-to-L6 synaptic resources | 0.8 / 1.2 s |

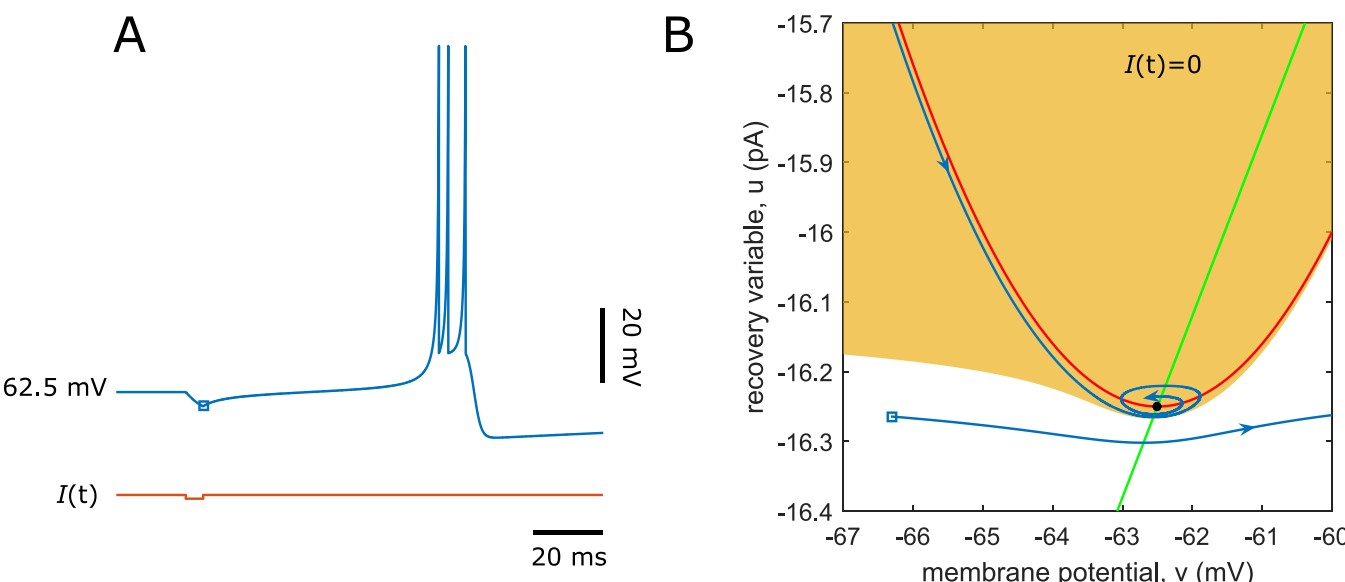

**Fig 8. Post-inhibitory bursts in a TC cell.** (A) A burst of spikes with significant latency is evoked by an inhibitory current pulse with 1 pA amplitude lasting 5 ms that approximates a transient inhibitory postsynaptic current of RE cells modulated by GABA$_A$ neurotransmitters. The membrane potential value immediately after the release from inhibition is denoted in the open square. (B) The phase portrait of the Izhikevich model with parameters used in the first subpopulation of TC cells when $I(t) = 0$. The intersection of the $v$-nullcline (red) and $u$-nullcline (green) is a stable focus, whose attraction basin is marked in orange. The trajectory initiated at the states right after the removal of inhibitory input (open square, same as in (A)) slowly goes through the neighborhood of the equilibrium before accelerating towards the voltage threshold of the spike peak. After setting off several spikes, the states are reset to the attraction domain and approach along the $v$-nuclline to the equilibrium (resting potential).

brief, weak negative current pulse pushes the states of a TC cell just outside its attraction basin but in the vicinity of the attractor, where the vector field is relatively small. The membrane potential increases slowly and steadily across the attractor neighborhood and fires a burst of spikes before entering the attraction domain (see Fig 8B). Hence the spike burst is induced with a long latency.

Furthermore, Fig 9 demonstrates how rhythmic bursting initiated by cortical feedback fades away in a thalamic circuit of 20 TC and 20 RE neurons. The TC cells that receive temporary excitatory corticothalamic inputs are driven to a depolarized level that prevents the post-inhibitory bursts. In contrast, most of the TC cells that have not received the corticothalamic feedback are capable of initiating the rebound burst in the first round of oscillation, induced by the activity of RE cells after the onset of the corticothalamic stimulation. In the second bursting cycle, some of the TC cells that are silent in the first cycle fire, but the activity of their coupled RE cells are weaker than that in the first round. As a result, very few TC cells are triggered by the reduced RE activity in the third round, and after that, the inhibition from RE cells is not strong enough to elicit any rebound burst in the TC cells, which terminates the rhythmic bursting.

The rhythmic activity initiated in the stimulated barreloid of the VPM only induces inadequate cortical responses in the principal barrel and cannot propagate to the surrounding barrels due to the activation threshold of the cortical neurons.

## Model robustness

Our model employed synaptic depression as a surrogate for inhibitory neurons to prevent overexcitation in the cortex. Fig 10B and 10C shows the sensitivity of SSA in cortical L4, L6

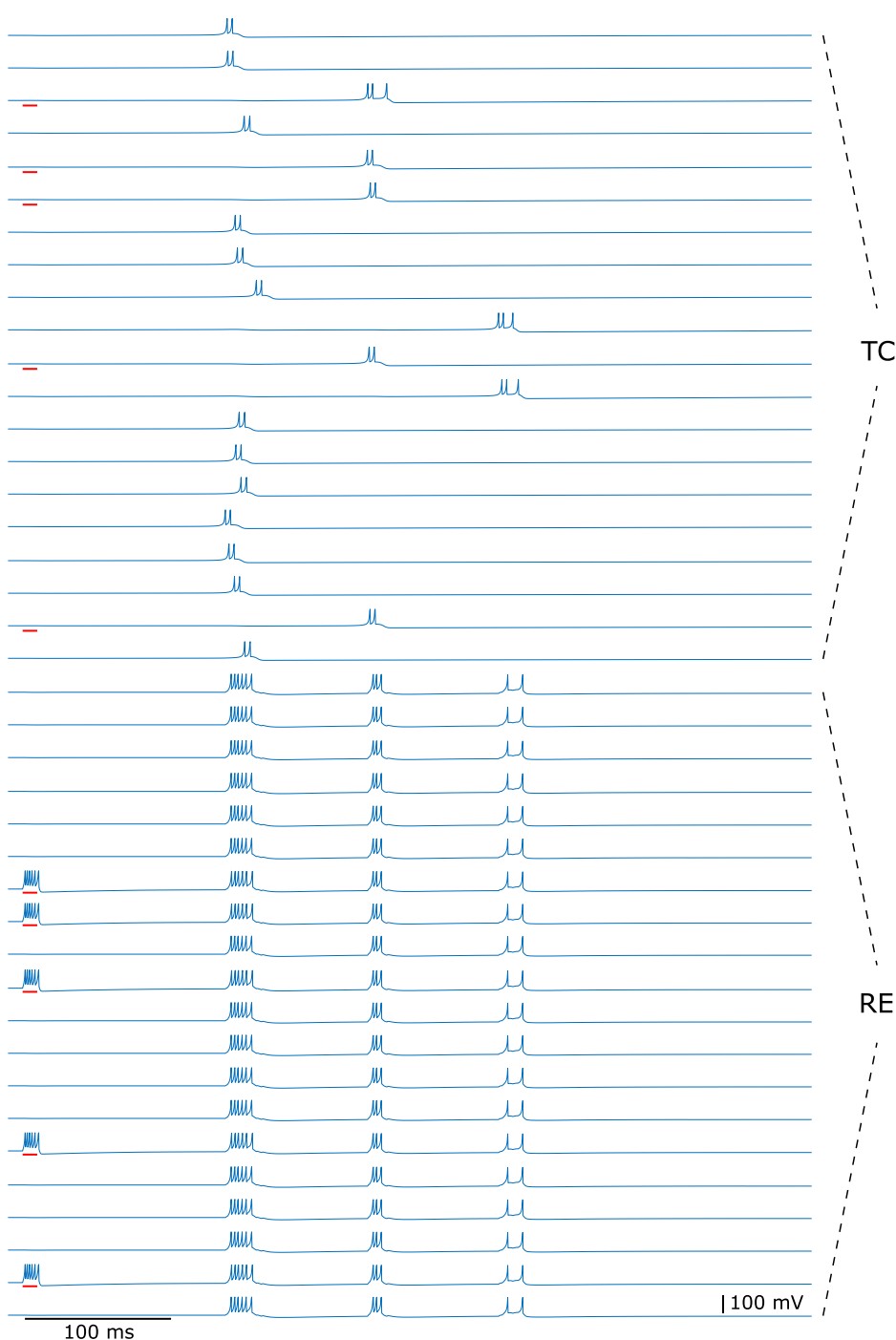

**Fig 9. Rhythmic bursting evoked by cortical discharges in a thalamic circuit of 20 TC and 20 RE cells.** Red bars under the voltage traces mark the initial cortical stimulation that is randomly distributed over 25% of TC and RE cells.

and thalamus to the recovery time constant of synaptic connections within cortical L4 and L6, respectively. The strength of SSA is quantified by the SSA index (SI). In particular, setting the synaptic resources to recover extremely fast (on the order of 1 ms as highlighted in the grey shade), the depression effects of the synapses become negligible and the cortical network

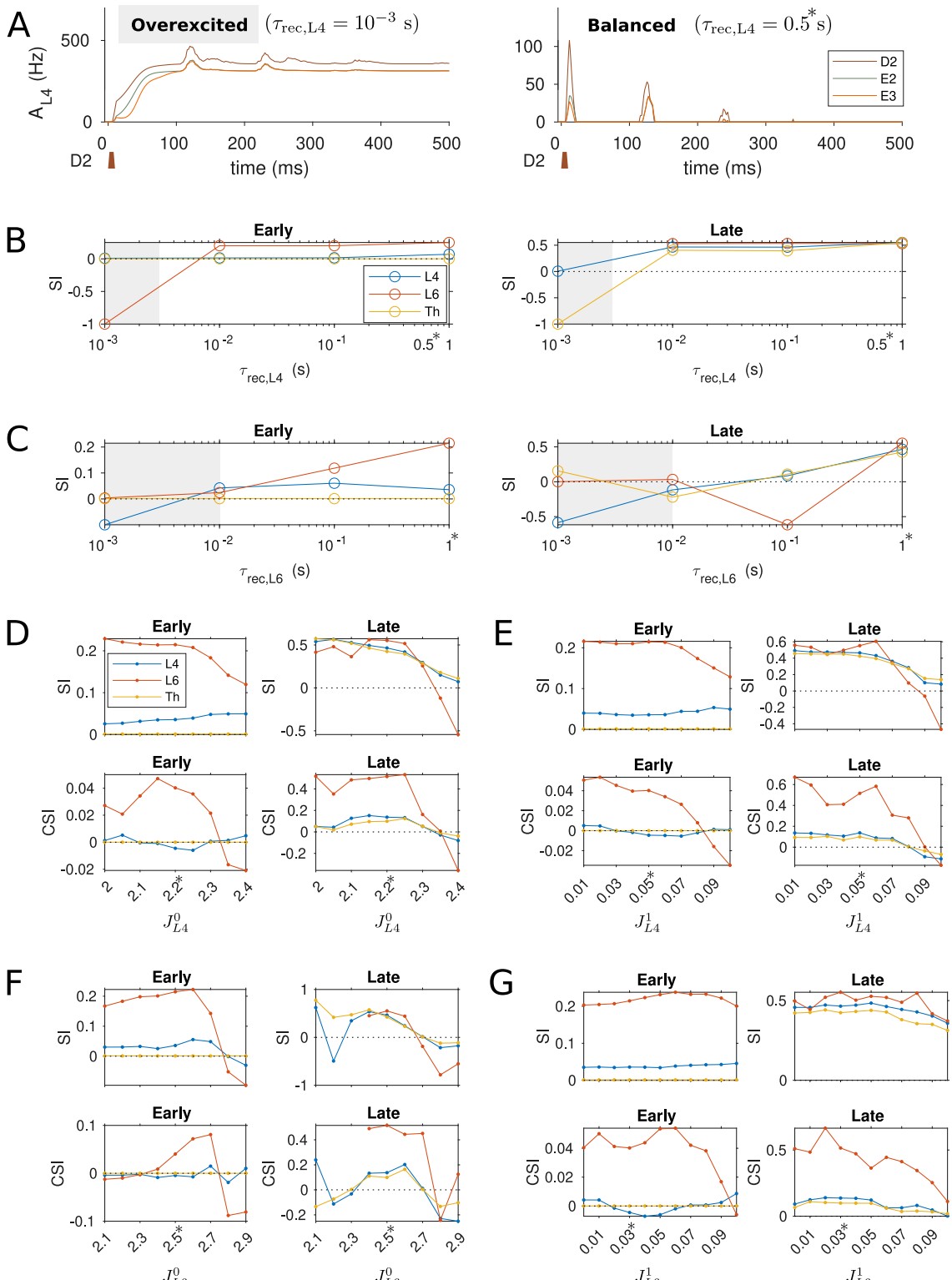

**Fig 10. Sensitivity of SSA and true deviance detection to the lateral cortical connections.** (A) Overexcited and balanced L4 population activity evoked by a brief D2 whisker deflection (marked under the time axis) given fast and slow recovery timescales of L4 depressing synapses, respectively. The starred value in the title denotes the parameter's default value. (B) Dependence of SSA in L4, L6 and thalamus on the logarithmically-scaled recovery time constant of L4 synapses, $\tau_{\mathrm{rec,L4}}$. The regimes for overexcited and balanced cortical activity are respectively represented in grey and white backgrounds. (C) Same as (B), but for L6 synapses, $\tau_{\mathrm{rec,L6}}$. (D-G)

Dependence of SSA and true deviance detection on the intra- and inter-columnar connection strengths within L4, $J_{L4}^0$, $J_{L4}^1$, and L6, $J_{L6}^0$, $J_{L6}^1$. All responses fall into the regime of balanced activity when modifying these lateral and recurrent connections. The starred values under the horizontal axes represent the parameters' default values. The random seed of model simulation is kept fixed when changing the values of each connection parameter.

generates epileptic-like behavior in response to a transient peripheral stimulus (depicted in the left panel of Fig 10A). In contrast, balanced cortical activity (right panel of Fig 10A) occurs when recovery of the depressing synapses is slower than 10 ms, and specifically, robust SSA can be obtained in the regime of hundreds of milliseconds to seconds of the synaptic recovery time constant.

As mentioned in the Discussion section, the generation of the deviant signal is sensitive to the extent of the lateral spread of activity across barrels in L4 and L6, which is largely confined by recurrent and lateral coupling strength within and between barrels. Therefore, we investigated the dependence of both SSA and true deviance detection on the intra- and inter-columnar connection strength within L4 and within L6 by varying one parameter at a time and fixing the rest to their given values, as demonstrated in Fig 10D–10G. No epileptic-like activity is able to be induced in these scenarios. The more positive value of the context-specificity index (CSI) indicates the stronger effect of true deviance detection. For lateral connections (right panel of Fig 10E and 10G), deviance-detecting effects no longer exist when their strength exceeds a certain critical value (0.07 for $J_{L4}^1$ and 0.08 for $J_{L6}^1$). On the other hand, the operating ranges of the recurrent connections where the model demonstrates qualitatively similar deviant behaviors to the experimental observations are 2.1 to 2.3 for $J_{L4}^0$ and 2.5 to 2.6 for $J_{L6}^0$. The relatively lower ratio of the operating range of recurrent connections to their given values indicates that the deviance-detecting effects in our model are more susceptible to recurrent than lateral cortical connections.

In addition, we studied the sensitivity of SSA and true deviance detection to the cortico-thalamic and intra-thalamic connection strength, as well as regimes of different thalamic late activity. Given different combinations of cortico-thalamic and intra-thalamic coupling strength, the thalamic circuit can trigger three late response patterns (see Fig 11A), which are also reflected in cortical L4 due to the thalamocortical pathway. As can be seen in Fig 11B–11E, the deviance-detecting effect is more robust in the decaying oscillation regime than in the other two regimes. Particularly, we found that as the TC-to-RE synaptic conductances increases, the regime of no oscillation shrinks while the regime of sustained oscillation expands (Fig 11B–11D). This transition of regimes is likely because the RE cells become more excitable by the increased conductances from TC cells onto them and, in turn, provide more inhibition to TC cells. Consequently, more rebound bursts are triggered in TC cells, which repeats the cycle of firings of RE cells. Finally, we fixed the cortico-thalamic connection strength to its default value and examined the sensitivity of the results to the intra-thalamic coupling strength (Fig 11E). As the TC-to-RE synaptic conductance increases, the response regime shifts from decaying oscillation to sustained oscillation, which weakens and even eliminates the deviance-detecting effect in the late responses.

## Data analysis

The average response to stimuli of a whisker identity is evaluated as mean spike counts of the stimulated column induced by all presentations of the whisker identity in all repeated protocols. The average spike counts are calculated as the integrals of population activity with a temporal resolution of 0.1 ms over a time window after the stimulus onset.

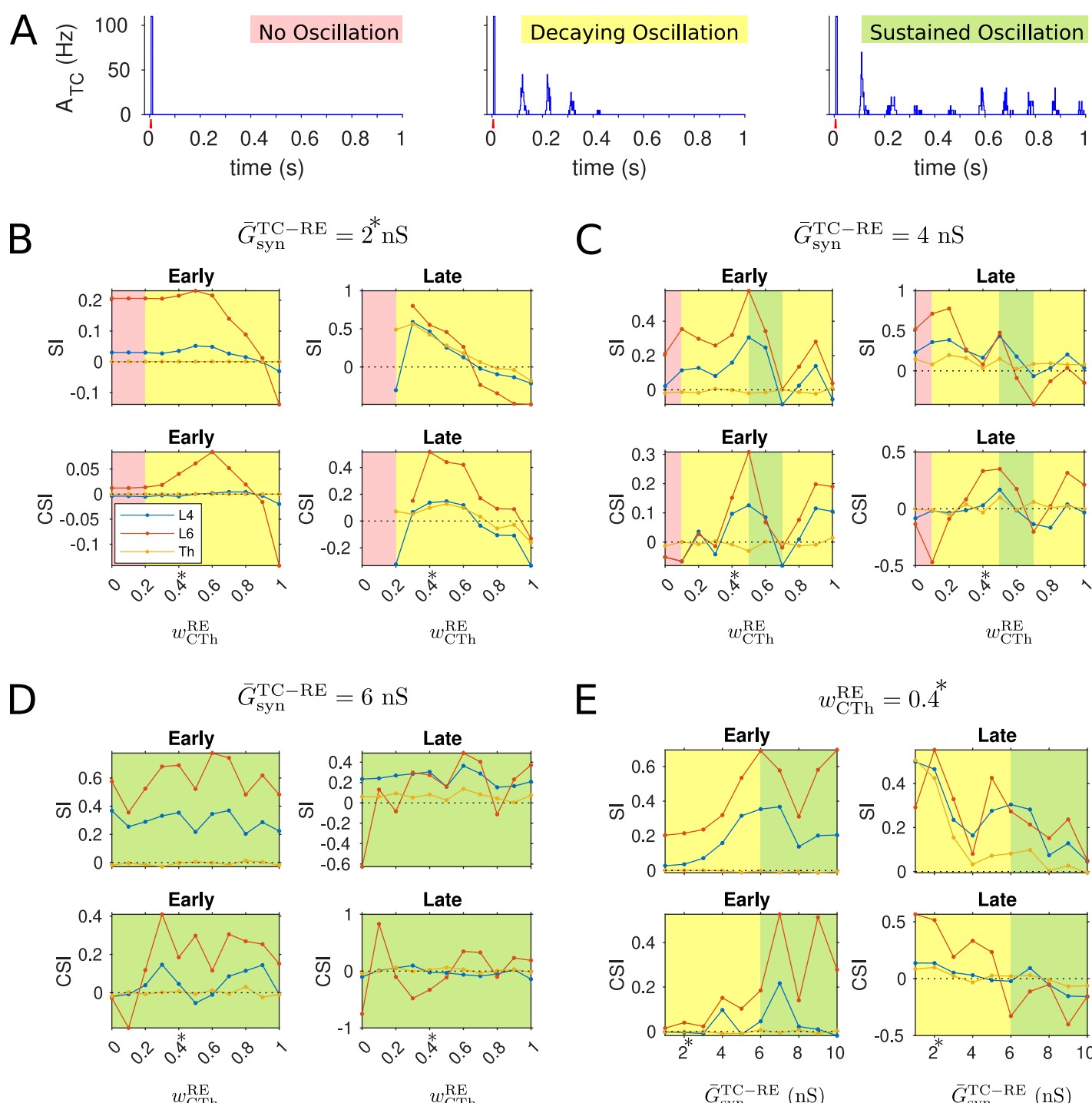

**Fig 11. Dependence of deviance-detecting effect and thalamic response patterns on the cortico-thalamic and intra-thalamic connection strength.** (A) Three late oscillation patterns (regimes) of thalamocortical cells can be elicited by a whisker deflection (marked in red under the time axis). (B-D) Sensitivity of SSA and true deviance detection in L4, L6 and thalamus to the strength of cortico-thalamic coupling on RE cells, $w_{CTh}^{RE}$, given different TC-to-RE synaptic conductances, $\bar{G}_{syn}^{TC-RE}$. Background colors denote one of the three regimes depicted in (A). (E) Analogous to (B-D), but by changing TC-to-RE synaptic conductances, $\bar{G}_{syn}^{TC-RE}$, with cortico-thalamic coupling strength, $w_{CTh}^{RE}$, fixed. The starred value denotes the parameter's default value.

The effect of SSA is quantified using the SSA index (SI), which is a normalized difference between the response to the deviant and the standard in the oddball protocol:

$$\text{SI} = \frac{\text{d}(x_1) - \text{s}(x_2)}{\text{d}(x_1) + \text{s}(x_2)} \qquad (33)$$

where $\text{d}(x_1)$ and $\text{s}(x_2)$ are the average responses measured as spike counts to the whisker identities $x_1$ and $x_2$ presented as the deviant and standard, respectively. Here, the roles of two whiskers as the deviant and standard are not swapped as our network is homogenous and symmetric, and consequently, there is no response bias towards a specific whisker identity. According to the formula, SI is normalized between -1 and 1 and its positive value indicates the existence of SSA, where stronger SSA effects lead to values of SI closer to 1.

Furthermore, to quantify the neuronal sensitivity to certain repeated patterns in its presentation context, we defined a context-specificity index (CSI) to compare the same deviant responses contextualized in regular and irregular sequences of stimuli:

$$\text{CSI} = \frac{\text{d}_{\text{reg}}(x) - \text{d}_{\text{irr}}(x)}{\text{d}_{\text{reg}}(x) + \text{d}_{\text{irr}}(x)} \qquad (34)$$

where $\text{d}_{\text{reg}}(x)$ and $\text{d}_{\text{irr}}(x)$ are the average spike counts to the same deviant whisker deflection $x$ respectively presented in regular and irregular paradigms, which can be either the pair of oddball and many-standards conditions, or sequenced and randomized conditions. Similar to SI, if the value of CSI is positive, then we conclude that the network model displays the capacity to identify underlying patterns from the input stream.

## Supporting information

**S1 Fig. Population activity of TC cells in response to deviant stimuli in oddball and many-standards conditions.** (A) Illustration of oddball (top) and many-standards (bottom) protocols. The deviant stimuli to the C2 whisker are highlighted in red in the oddball condition and green in the many-standards condition. (B) Temporal profiles of population activity of TC cells in C2 barreloid evoked by the oddball deviant (red) and many-standards deviant (green). (C) Magnified deviant responses in the shaded area denoted in (B). The stimulus duration is marked below the responses in the first row. (D) Average TC responses in C2 barreloid over inter-stimulus intervals to the oddball deviant (red) and many-standards deviant (green). (EPS)

**S2 Fig. Response patterns of TC cells in thalamic networks with different proportions of TC cells in the two complementary subgroups of each barreloid.** A brief D2 whisker deflection evokes different patterns of TC population activity $A_{\text{TC}}$ (averaged over all 100 TC neurons) in the principle barreloid, where the proportion of post-inhibitory bursting TC neurons to all TC neurons $P_{\text{TC}}$ increases (A: $P_{\text{TC}} = 0.2$, B: $P_{\text{TC}} = 0.4$, C: $P_{\text{TC}} = 0.6$ that is the one used in our model, D: $P_{\text{TC}} = 0.8$). The D2 whisker deflection stimulus is marked in red under the first panel. (EPS)

## Author Contributions

**Conceptualization:** Chao Han, Giacomo Indiveri, Wolfger von der Behrens, Eleni Vasilaki.

**Data curation:** Chao Han.

**Formal analysis:** Chao Han, Aditya Gilra.

**Funding acquisition:** Giacomo Indiveri, Wolfger von der Behrens, Eleni Vasilaki.

**Investigation:** Gwendolyn English, Wolfger von der Behrens.

**Methodology:** Chao Han.

**Project administration:** Giacomo Indiveri, Wolfger von der Behrens, Eleni Vasilaki.

**Resources:** Gwendolyn English, Wolfger von der Behrens.

**Software:** Chao Han.

**Supervision:** Hannes P. Saal, Aditya Gilra, Wolfger von der Behrens, Eleni Vasilaki.

**Validation:** Chao Han.

**Visualization:** Chao Han.

**Writing – original draft:** Chao Han, Wolfger von der Behrens.

**Writing – review & editing:** Chao Han, Gwendolyn English, Hannes P. Saal, Giacomo Indiveri, Aditya Gilra, Wolfger von der Behrens, Eleni Vasilaki.

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
