## [Decision Letter · Decision Letter 0]

30 Mar 2022

Dear Mr. Han,

Thank you very much for submitting your manuscript "Modelling novelty detection in the thalamocortical loop" for consideration at PLOS Computational Biology.

As with all papers reviewed by the journal, your manuscript was reviewed by members of the editorial board and by several independent reviewers. In light of the reviews (below this email), we would like to invite the resubmission of a significantly-revised version that takes into account the reviewers' comments.

We cannot make any decision about publication until we have seen the revised manuscript and your response to the reviewers' comments. Your revised manuscript is also likely to be sent to reviewers for further evaluation.

Sincerely,

Boris S. Gutkin

Associate Editor

PLOS Computational Biology

Wolfgang Einhäuser

Deputy Editor

PLOS Computational Biology

Reviewer's Responses to Questions

**Comments to the Authors:**

Reviewer #1: Han et al. built a multi-scale recurrent network based on a thalamocortical network model with synaptic depression to simulate the phenomenon of SSA and MMN by rendering adaptation property on those Layer 6 cortical neurons. Their model can explain the novelty detection mechanism in the thalamocortical loop. Meanwhile, this computational model is the first one to delineate the biphasic sensory response in the whisker-related part of S1. This paper shows a logical construct and clear experimental procedure to verify the hypothesis. However, whether other layers especially Layer 5 neurons contribute to the formation of cortical SSA is not incorporated in the current model, which cannot be neglected in the physiological condition. Therefore, some issues should be addressed (see below) before acceptance.

Main points

1. “we modelled each barreloid in the VPM nucleus by a cluster of 100 TC cells mediated by a pool of 100 RE cells,” May I ask if 100 as the population of cells is a reasonable parameter? Maybe you could cite a reference to explain your choice of population. In addition, I am interested in the effect of the different populations such as 10 or 1000 on your results. Similarly, I am confused if random stimulation to different proportions of TC cells in the two complementary subgroups of each barreloid with sensory input affects your experimental results. I am very pleased to see you try it.

2. In Fig 7 and Fig 8, I am very interested in the CSI of L6 in the late response, although I know the additional late responses can hardly spread into L6. I am very thanks for you if you show me that.

3. You maybe explain more about your hypothesis that L2/3 and L5 contribute little to the cortical SSA except citing these papers because this is the basis of your model architecture. For example, you could compare between excitation levels of different layers according to some biological references.

Minor:

1. Some claims should be properly supported by references, for example:

Line 197: “SSA is generally tested in the oddball protocol, where repetitive deflection is applied to 197 one whisker (called the standard), and the sequence is randomly interrupted from time 198 to time by deflection of another whisker (called the deviant).”

Line 497: “It has been observed that a cortico-thalamo-cortical loop structure exists in the rodent 497 whisker system.”

It is suggested that authors check the whole manuscript carefully.

Reviewer #2: The manuscript by Han et al. describes a model for mismatch negativity (MMN). The model consists of mean field models for pools of cortical neurons in layers 4 and 6 (L4 and L6) and simplified Izhikevitch models for interacting excitatory and inhibitory neurons. The key features of the model is that intracortical synapses allow the vertical spread of activity from L4 to L6 and the lateral spread of activity across barrels in L4 and L6. L6 activity recruits thalamic spindle oscillations which give rise to late rhythmic responses in the cortex. The basic idea of the model is that due to synaptic depression of L4 to L6 projections, repeated stimuli do not evoke enough L6 activity to recruit thalamic spindle oscillations (via strong corticothalamic feedback to reticular neurons). A deviant stimulus is able to recruit these spindles due to absence of depression. In a many standards condition, there is less depression of thalamocortical synapses in the control pathway, leading to greater L4 activity and lateral spread of activity, which leads to some synaptic depression that prevents effective recruitment of layer 6 and thalamic spindles.

I find the basic idea (that MMN is driven by synaptic depression limiting the interlaminar spread of activity which in turn limits the recruitment of thalamic spindle activity that could drive late rhythmic responses) potentially interesting. However, I think there are some questions about the robustness of these results (see comment #2), their relationship to what is known about the biology (see comments #1 and 3), which specific aspects of the model are actually necessary (see comment #1), and some apparent inconsistencies (see #4).

1. I’m a bit confused about the many-standards condition for two reasons. First, does the lateral spread of activity occur mainly via synapses in L4 or L6? It seems like it is in L4 based on Fig. 5B-C. If you had spread of activity mainly through L6 synapses, then you wouldn’t get L4 to L6 depression, so this wouldn’t work right? So doesn’t the model make a strong prediction that MMN depends on lateral connections before L6 (most likely in L2/3)? Does this mean you could you dispense with L6 connections and have the model work just as well? And how well does this line up with the anatomy – my understanding is that there are many more lateral connections in L2/3 (and L5) than L4 or L6. This should be discussed.

2. A second related issue is about the parameter space and robustness of the model. It seems like the model has been finely tuned so that in the many standards condition activity spreads a bit to other columns and activates their L4->L6 synapses just enough to produce some depression but not so much that you get extensive recruitment of thalamic spindles in these pathways. How robust is this, i.e., can you show some measure of the parameter space and how much you can enhance / weaken the strength of the lateral connections and still get this effect?

3. Based on the model, it seems like there should be more robust late responses (corresponding to weak spindle-driven activity) in response to the standard stimulus in the many-standards condition compared to the single-standard (‘oddball) condition. How well does this line up with experimental observations?

4. I’m confused by comparing Figs. 4C and 5E. In 4C there is similar late TC activity for the oddball deviant and many-standards deviant. But in 5E there is almost no late TC activity for the many standards deviant. What explains this discrepancy?

5. Overall I found the figures difficult to decipher. Lots of traces are shown, in many cases different traces corresponding to different things in different panels. This is particularly true for Figure 5 – some panels compare C2 traces for x_L4 whereas others compare D2 traces for x_L6. It would be easier if the same thing was shown consistently, with arrows and boxes to highlight the key points. As it is, it is very hard to keep track of what is going on.

Reviewer #3: Stimulus specific adaptation (SSA) is a phenomenon where responses to deviant stimuli are larger than the responses to non-deviant stimuli. The effect was mostly described in the auditory cortex, but was found also in the whisker somatosensory cortex. The manuscript describes a mechanistic explanation to the effect: neurons in cortical layer 6 adapt, via synaptic depression, specifically to a frequently presented stimulus, resulting in reduced population activity when compared with the population activity evoked by a rare stimulus. This difference is projected from cortex to thalamus and generate decaying spindle oscillations. As a result, layer 4 neuron in cortex display late responses to rare stimuli.

The mechanism presented in the model is interesting, but suffers from substantial shortcomings. Spindle oscillations are generated during sleep or anesthesia, when thalamic neurons are hyperpolarized and function in a "bursting mode", during which they possess post-inhibitory rebound and fire a calcium spike when released from hyperpolarization. In awake animals, however, bursting model is rarely seen and thalamic neurons fire tonically in response to excitatory stimulation. Therefore, a mechanism based on activity during sleep cannot explain sensory responses in behaving animals.

Thalamic neurons in the model burst in response to touch (Fig. 2D). In reality, they fire, on average, less than one spike per touch (0.6 spikes/touch; Yu et al., Nat. Naurosci. 12:1647, 2016). The thalamic model display decaying spindle oscillations. To achieve them, parameters of thalamic connectivity should be fine-tuned. Otherwise, one will not obtain any oscillations or obtained sustained oscillations. The model does not include excitatory neurons in cortex. Without them, epileptic-like pulses propagate in cortex. It is not clear what stop them in the model.

The authors emphasize the small subset of neurons in the granular layer with a deviant-specific late response, occurring roughly 200 ms after stimulus offset, found experimentally in ref. 8. These neurons display late response even to non-deviant stimuli (Fig. 4B), although less oscillatory than the response to the deviant stimulus. In the model, there is late response in thalamus even for a standard stimulus (Fig. 4C), that is also weaker than the response for the deviant stimulus. This means that "spindle-like" oscillations are generated even in response to standard stimuli. In reality, rodent whisk in about 10 Hz (rats) – 15 Hz (mice). This means that responses to subsequent touch events will be "polluted" by oscillations generated in response to the first touch. This will make the responses to consecutive touch events complicated, and, as far as I know, has never been reported.

Fig. 6 shows that the response to sequenced deviant is almost the same as the response to randomize deviant. I do not understand what the authors tried to say by presenting this figure, and why they emphasized the tiny difference between the two responses.

For all these reasons, I find that the model does not describe the real biological system well.

**Have the authors made all data and (if applicable) computational code underlying the findings in their manuscript fully available?**

Reviewer #1: Yes

Reviewer #2: Yes

Reviewer #3: Yes

PLOS authors have the option to publish the peer review history of their article (what does this mean?). If published, this will include your full peer review and any attached files.

Reviewer #1: No

Reviewer #2: No

Reviewer #3: No
---

## [Decision Letter · Decision Letter 1]

28 Aug 2022

Dear Mr. Han,

Thank you very much for submitting your manuscript "Modelling novelty detection in the thalamocortical loop" for consideration at PLOS Computational Biology.

As with all papers reviewed by the journal, your manuscript was reviewed by members of the editorial board and by several independent reviewers. In light of the reviews (below this email), we are unable to accept the paper in its present form. However you are invited to  resubmit a significantly-revised version that takes into account the reviewers' comments.

Notably, the questions from reviewer 1 must be answered clearly and conclusively. This reviewer questions how realistic are the extensive L6 connections  - is there conclusive evidence for them in the cortex? Second question is about results presented in figure 4C and 5D - these issues must be cleared up.

Issues raised by reviewer 2 are also quite serious and must be addressed fully. Notably - the reviewer questions how realistic is the TC bursting mode in vivo non-anesthetized state? Seems like this is a crucial point for the validity of the of the model.  Realism of the model without cortical inhibitory neurons must be also cleared up - notably the regimes for the SSA and run-away excitation. Furthermore, authors must address the doubts on the robustness of the model w.r.t. parameter choices - e.g. the strength thalamus-cortex coupling etc.

We cannot make any decision about publication until we have seen the revised manuscript and your response to the reviewers' comments. Your revised manuscript is also likely to be sent to reviewers for further evaluation.

Sincerely,

Boris S. Gutkin

Academic Editor

PLOS Computational Biology

Wolfgang Einhäuser

Section Editor

PLOS Computational Biology

Reviewer's Responses to Questions

**Comments to the Authors:**

Reviewer #2: In my original comment 1, I asked “does the lateral spread of activity occur mainly via synapses in L4 or L6… doesn’t the model make a strong prediction that MMN depends on lateral connections before L6?” I also asked about how this lines up with what is known about the existence of lateral connections in different layers. In response, the authors state that the depression of intra-column L6 synapses is the primary cause of deviant-selectivity. They go on to state that the recovery of synaptic resources in L4 means that the input to L6 is similar in the many standards and oddball deviant conditions, resulting in similar L6 activation in both of these cases (and weaker activation in the oddball standard condition). They then state that there is sometimes propagation into the deviant column in the many standards condition but not in the oddball standard condition, which causes depletion of L6 resources in the many standards case. Based on this, it sounds like the model requires lateral connections in L6 to produce a difference between the many standards and oddball deviant conditions. However, I am still unclear on how realistic this is – there are known to be extensive lateral connections in L2/3 and L5, but is this the case for L6, as the model seems to require.

In my original comment 4 I asked about the reason for the difference between Fig. 4C (in which the late response to the many standards deviant is similar to the late response to the oddball deviant) and 5D (in which there is very little late response to the many standards deviant). The authors respond by stating that Fig. 4C is an average whereas Fig. 5D is a ‘typical example.’ But it seems that the example in Figure 5D is not typical of the average response shown in Fig. 4C? Furthermore the average late response of the model to the many standards deviant shown in Fig. 4C seems very different from the experimentally measured late response to the many standards deviant shown in the bottom row of Fig. 4B (in which the late response to the many standards deviant is very weak making it different from Fig 4C but also has a different timecourse compared Fig. 5C). These discrepancies strike me as problematic and must be clearly addressed.

Reviewer #3: The authors responded to my previous comments. A major source of confusion is the lack of distinction between healthy, awake behavior and cases like anesthesia, absence epilepsy and sleep. Generally, thalamic neurons function in the spiking mode during awake behavior, although there may be certain cases in which some neurons – but not the majority – respond in a burst to stimuli. During some forms of anesthesia, weak sleep and (maybe) absence seizure, the bursting mode is abundant. The authors present SSA as a property of sensory systems, and relate it to mismatch negativity (MMN) that is observed in humans. In contrast, the authors explain SSA using a mechanism based post-inhibitory rebound in TC neurons, which is a hallmark of anesthetized, unconscious states. Post-inhibitory rebound occurs when a TC neuron is hyperpolarized, such as during sleep or anesthesia. For this reason, the authors obtain periodic activity in the frequency of about 10 Hz in response to some stimuli (Fig. 3). They present experimental results from Musall et al. (Fig. 4B), but these experiments were carried out on anesthetized animals.

Whether or not a model that is valid for anesthetized animals and aims to explain sensation is interesting or not is a matter of editorial decision. If this manuscript is published, the distinction between awake and anesthetized states should be clearly written. In addition to this issue, there are several other shortcomings in the article.

Comments 3.3: The model does not include inhibitory neurons in cortex. Without such neurons, epileptic-like activity is expected to propagate in cortex due to the excitatory-to-excitatory connections within each layer. The authors did not respond to this criticism, and discuss properties of TC neurons instead. The authors need to examine under which conditions such epileptic-like state exists by examining the connections within later 4 and within layer 6. They need to clarify in which regime does they obtain SSA but no epileptic-like activity in cortex.

Comment 3.4: The authors admit that: "Indeed, parameters of thalamic neurons and their connectivity strength need to be finely tuned to demonstrate the decaying rhythmic bursting behavior". It is therefore unclear whether the model is robust. The authors need to examine the robustness of their results by modifying the strength of the thalamocortical or cortico-thalamic connections. They need to report in which regime decaying oscillations are obtained (such as in Fig. 3), in which regime the oscillations continue "forever" and do not decay, and in which regime there are no oscillations.

Comment 3.5: The response to sequenced deviant is almost the same as the response to randomize deviant. The authors have not convinced that the small difference is not a result of a particular choice of parameters. Figure 6 should be removed.

**Have the authors made all data and (if applicable) computational code underlying the findings in their manuscript fully available?**

Reviewer #2: Yes

Reviewer #3: Yes

PLOS authors have the option to publish the peer review history of their article (what does this mean?). If published, this will include your full peer review and any attached files.

Reviewer #2: No

Reviewer #3: No
---

## [Decision Letter · Decision Letter 2]

8 Jan 2023

Dear Mr. Han,

Thank you very much for submitting your manuscript "Modelling novelty detection in the thalamocortical loop" for consideration at PLOS Computational Biology. As with all papers reviewed by the journal, your manuscript was reviewed by members of the editorial board and by several independent reviewers. The reviewers appreciated the attention to an important topic. Based on the reviews, we are likely to accept this manuscript for publication, providing that you modify the manuscript according to the review recommendations.

Sincerely,

Boris S. Gutkin

Academic Editor

PLOS Computational Biology

Wolfgang Einhäuser

Section Editor

PLOS Computational Biology

Reviewer's Responses to Questions

**Comments to the Authors:**

Reviewer #2: The authors have responded satisfactorily to my concerns.

Reviewer #3: In the revised version, the model is presented as a model for anaesthetized somatosensory cortex. Whether this makes it less interesting, the existence of thalamic bursts is easier to justify based on experimental observations. Since the system can be less active for certain forms of anesthesia, it is more justified to neglect inhibition in cortex.

The authors need to remove the remaining of Fig. 6. After panel 6B was removed, the remaining panel does not carry enough relevant information to be show.

**Have the authors made all data and (if applicable) computational code underlying the findings in their manuscript fully available?**

Reviewer #2: None

Reviewer #3: Yes

PLOS authors have the option to publish the peer review history of their article (what does this mean?). If published, this will include your full peer review and any attached files.

Reviewer #2: No

Reviewer #3: No

Figure Files:

Data Requirements:

Reproducibility:

References:

---

## [Editor Report · Decision Letter 3]

21 Feb 2023

Dear Mr. Han,

We are pleased to inform you that your manuscript 'Modelling novelty detection in the thalamocortical loop' has been provisionally accepted for publication in PLOS Computational Biology.

Best regards,

Boris S. Gutkin

Academic Editor

PLOS Computational Biology

Wolfgang Einhäuser

Section Editor

PLOS Computational Biology

---

## [Editor Report · Acceptance letter]

27 Mar 2023

PCOMPBIOL-D-21-02007R3 

Modelling novelty detection in the thalamocortical loop

Dear Dr Han,

I am pleased to inform you that your manuscript has been formally accepted for publication in PLOS Computational Biology. Your manuscript is now with our production department and you will be notified of the publication date in due course.

With kind regards,

Zsofi Zombor
